# Don't Pass@k: A Bayesian Framework for Large Language Model Evaluation

**Mohsen Hariri,**[*] **Amirhossein Samandar,**[†] **Michael Hinczewski,**[†] **Vipin Chaudhary**[*]

[*]Department of Computer and Data Sciences, Case Western Reserve University, Cleveland, OH, USA

[†]Department of Physics, Case Western Reserve University, Cleveland, OH, USA

`{mohsen.hariri, amirhossein.samandar, mxh605, vipin}@case.edu`

## Abstract

Pass@$k$ is widely used to report the reasoning performance of LLMs, but it often produces unstable and potentially misleading rankings, especially when the number of trials (samples) is limited and computational resources are constrained. We present a principled Bayesian evaluation framework that replaces Pass@$k$ and average accuracy over $N$ trials (avg@$N$) with posterior estimates of a model's underlying success probability and credible intervals, yielding stable rankings and a transparent decision rule for differences. Evaluation outcomes are modeled as categorical (not just 0/1) with a Dirichlet prior, giving closed-form expressions for the posterior mean and uncertainty of any weighted rubric and enabling the use of prior evidence when appropriate. Theoretically, under a uniform prior, the Bayesian posterior mean is order-equivalent to average accuracy (Pass@1), explaining its empirical robustness while adding principled uncertainty. Empirically, in simulations with known ground-truth success rates and on AIME'24/'25, HMMT'25, and BrUMO'25, the posterior-based procedure achieves faster convergence and greater rank stability than Pass@$k$ and recent variants, enabling reliable comparisons at far smaller sample counts. The framework clarifies when observed gaps are statistically meaningful (non-overlapping credible intervals) versus noise, and it naturally extends to graded, rubric-based evaluations. Together, these results recommend replacing Pass@$k$ for LLM evaluation and ranking with a posterior-based, compute-efficient protocol that unifies binary and non-binary evaluation while making uncertainty explicit. Source code is available at **Scorio**[1].

## 1 Introduction

Large language models (LLMs) have moved rapidly from research artifacts to everyday infrastructure (1; 2). Students use them for homework and exam preparation; developers rely on them for code synthesis and refactoring (3); analysts and clinicians use them for decision support; and agents built atop LLMs are increasingly embedded in workflows across industry and government. As deployment and investment accelerate (4), trust, oversight, and comparability become central: *how we evaluate LLMs* directly shapes which models are adopted, what progress is declared, and how resources are allocated (5; 6; 7; 8; 9; 10; 11).

Evaluation, however, remains the weakest link in the LLM pipeline. Alongside advances in model efficiency and compression(12; 13; 14; 15; 16; 17; 18; 19), training and fine-tuning methods such as parameter-efficient fine-tuning (PEFT), low-rank adaptation (LoRA), and reinforcement learning from human feedback (RLHF) (20; 21; 11), and inference/decoding (sampling strategies, caching, efficient attention) (22; 23), the community still leans on simple, yet flawed, success rates and Pass@$k$-style metrics to summarize capabilities (24). These practices are convenient but fragile. On small or costly benchmarks (e.g., math reasoning sets with only tens of problems such as AIME) (25; 26), Pass@$k$ or single-run accuracy often produce unstable rankings (27), are sensitive to decoding choices and seed effects (22; 28), and provide little guidance on whether observed gaps are

---

[1]`https://github.com/mohsenhariri/scorio`. See Appendix G for API documentation.

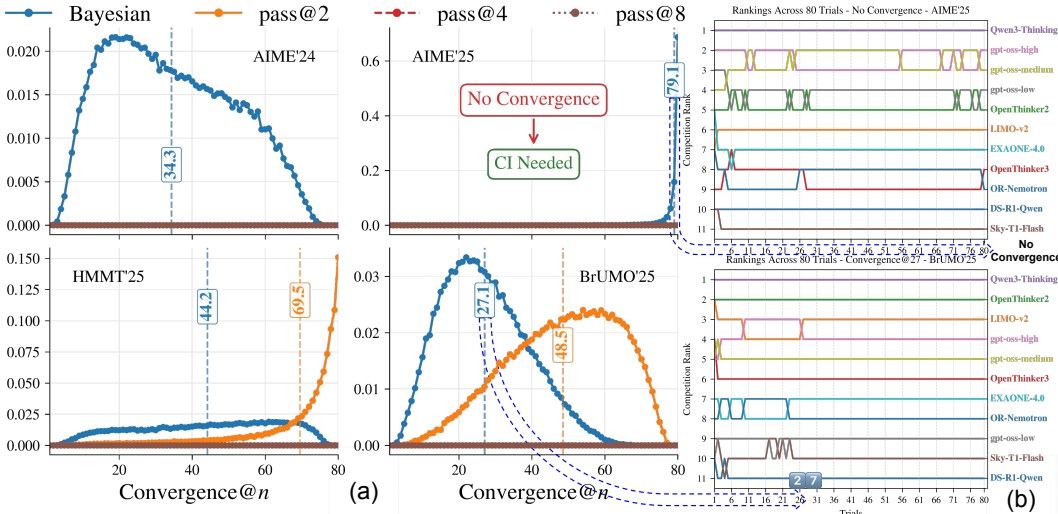

Figure 1: a) Probability mass functions (PMFs) of convergence@$n$, the number of trials $n$ above which a ranking of LLM models consistently matches the ranking using $N_{max} = 80$ trials. Eleven LLM models (listed on the right) and four math-reasoning datasets are used—AIME'24, AIME'25, HMMT'25, and BrUMO'25—comparing Pass@2/4/8 against our Bayesian posterior evaluation (Bayes@$N$). Each PMF is estimated by bootstrapping with $10^5$ samples over the $N_{max}$ trials; vertical lines indicate the mean of each convergence distribution. On AIME'24/'25, the Pass family frequently *fails to converge*, whereas Bayes@$N$ converges. On HMMT and BrUMO, Pass methods converge more slowly (mean required trials $\approx$ 69.5 and $\approx$ 48.5) than Bayes@$N$ ($\approx$ 44.2 and $\approx$ 27.1), respectively. Right: Example competition-style ranking from a single bootstrap replicate, highlighting the mean convergence for AIME'25 and BrUMO'25. Per task rankings, including worst-case replicates, are in Appendix J (Figure 10).

meaningful or mere noise (29; 30). Averaging across multiple runs ("avg@$N$") helps but is compute-hungry (31), offers no unified way to handle graded/rubric outcomes, and lacks a principled decision rule for significance (29; 32; 33).

This paper takes a different approach: we treat evaluation itself as a statistical inference problem. We introduce a *posterior-based* framework that replaces Pass@$k$ and avg@$N$ with estimates of a model's underlying success probabilities and associated uncertainty (34). Outcomes are modeled as *categorical* (35) rather than purely binary: each item can yield correct, partially correct, formatting-error, refusal, or rubric-defined levels. A Dirichlet prior over these categories yields closed-form posterior means and credible intervals for any *weighted rubric*, allowing the evaluator to report both a point estimate and principled uncertainty with negligible overhead. In the binary special case under a uniform prior, its posterior mean is order-equivalent to average accuracy, explaining the empirical robustness of avg@$N$ while making uncertainty explicit.

The framework addresses *four* persistent pain points. ❶ *Convergence*: as shown in Figure 1, methods differ substantially in how many trials they need to reach a stable ranking. ❷ *Credible intervals*: a simple rule—**do not declare a winner when intervals overlap**—reduces leaderboard churn and over-interpretation of tiny gaps. Analytic updates let evaluators monitor interval widths online and allocate extra trials only when needed. ❸ *Categorical evaluation*: graded rubrics are natural in this framework, covering step-by-step reasoning, partial credit, or judge categories without ad hoc aggregation. ❹ *Prior information*: we can incorporate prior evidence when appropriate (e.g., reuse of stable rubric distributions across closely related tasks or versions).

We validate the approach in two settings: In controlled simulations with known ground-truth success rates, the posterior procedure converges to correct rankings with fewer samples than Pass@$k$ and recent variants, and it flags when ties are statistically unresolved. On real math-reasoning benchmarks (AIME'24/'25 (25; 26), HMMT'25 (36), and BrUMO'25 (37)-derived sets), we observe the same pattern: the posterior method achieves greater rank stability at far smaller sample counts than Pass@$k$, while clarifying when differences are meaningful versus noise. Practically, this yields a computationally efficient protocol that is easy to implement and audit.

We summarize our contributions as follows:

- **A unified Bayesian evaluation framework.** We model per-item outcomes as categorical with a Dirichlet prior, yielding closed-form posterior means and credible intervals for *any* weighted rubric, with binary evaluation as a special case. This unifies 0/1 and graded evaluations and supports reuse of prior evidence when justified.
- **A compute-efficient, interval-aware protocol.** We provide a simple recipe: report posterior means with credible intervals; only declare differences when intervals do not overlap; adaptively allocate additional samples until intervals meet pre-specified widths. This protocol naturally supports sequential/online evaluation.
- **Empirical evidence on simulations and math benchmarks.** On synthetic data with known ground truth and on AIME'24/'25, HMMT'25, and BrUMO'25 datasets, our method achieves faster convergence and greater rank stability than Pass@$k$ and recent variants, enabling reliable comparisons with far fewer samples.

## 2 BAYESIAN FRAMEWORK FOR EVALUATING LLM PERFORMANCE

### 2.1 BACKGROUND: THE PASS@$k$ METRIC AND ITS LIMITATIONS

Evaluation metrics for LLMs aim to quantify performance on tasks like reasoning or programming, but they often struggle to provide reliable relative rankings across models. Pass@$k$, for instance, estimates the probability of at least one correct answer within $k$ model attempts (see Appendix H for details). While convenient, this metric exhibits high variance (38), particularly when $k$ approaches the total number of trials, $N$, resulting in unstable rankings (24). Small fluctuations in correctness can distort comparisons, particularly in benchmarks with few problems or limited computational resources, raising doubts about its suitability for differentiating model capabilities. If a metric cannot consistently distinguish stronger models from weaker ones, its value as a benchmarking tool is undermined (27).

Estimating uncertainty in Pass@$k$ scores is also challenging, as it lacks closed-form expressions for variance, relying instead on computationally intensive approximations like bootstrapping. A truly effective metric should yield reliable performance rankings with a minimal number of trials, prioritizing both accuracy and efficiency in resource-constrained environments. To address these limitations, we propose a Bayesian evaluation framework that provides more stable estimates of performance, incorporates uncertainty, and facilitates robust relative comparisons across models (34; 39; 40).

### 2.2 RESULTS MATRIX

Consider a results matrix $R$ for an LLM evaluated on a test set comprising $M$ questions. Due to the stochastic nature of LLM sampling, responses may vary across independent trials, so we run the LLM $N$ times per question. The outcomes are captured in the $M \times N$ matrix $R$, where element $R_{\alpha i}$ represents the score in the $i$th trial for the $\alpha$th question. This score is an integer ranging from 0 to a maximum value $C$, reflecting a rating system with $C + 1$ categories. In the binary case ($C = 1$), 0 indicates an incorrect answer and 1 a correct one, though we accommodate more nuanced rubrics generally.

### 2.3 WEIGHTED PERFORMANCE METRIC

For the $\alpha$th question, $\alpha = 1, \ldots, M$, there is an underlying probability $\pi_{\alpha k}$ that the LLM's answer falls in the $k$th category. We denote $\boldsymbol{\pi}_\alpha$ as the $(C + 1)$-dimensional vector with elements $\pi_{\alpha k}$, $k = 0, \ldots, C$. If all $\boldsymbol{\pi}_\alpha$ were known, we could calculate a desired performance metric $\bar{\pi}$ as a weighted average over these probabilities:

$$\bar{\pi} = \frac{1}{M} \sum_{\alpha=1}^{M} \boldsymbol{w} \cdot \boldsymbol{\pi}_\alpha = \frac{1}{M} \sum_{\alpha=1}^{M} \sum_{k=0}^{C} w_k \pi_{\alpha k}, \tag{1}$$

---

**Algorithm 1** LLM performance evaluation using the Bayes@$N$ framework.

---

**function** EVALUATEPERFORMANCE($R$, $[R^0]$, $\boldsymbol{w}$)
    **input:** $M \times N$ matrix $R$ of results, with each element $R_{\alpha i} = 0, \ldots, C$
            weight vector $\boldsymbol{w} = (w_0, \ldots, w_C)$ defining performance metric $\bar{\pi}$
    **optional input:** $M \times D$ matrix $R^0$ of results for prior; otherwise $D = 0$
    **output:** performance metric estimate $\mu$ and associated uncertainty $\sigma$

    $T = 1 + C + D + N$
    **for** $\alpha = 1$ to $M$ **do**                                           ▷ Tally results in $R$ and $R^0$
        **for** $k = 0$ to $C$ **do**
             $n_{\alpha k} = \sum_{i=1}^{N} \delta_{k, R_{\alpha i}}$
             $n_{\alpha k}^0 = 1 + \sum_{i=1}^{D} \delta_{k, R_{\alpha i}^0}$
             $\nu_{\alpha k} = n_{\alpha k}^0 + n_{\alpha k}$
        **end for**
    **end for**
    $\mu = w_0 + \frac{1}{MT} \sum_{\alpha=1}^{M} \sum_{j=0}^{C} \nu_{\alpha j}(w_j - w_0)$
    $\sigma = \left[ \frac{1}{M^2(T+1)} \sum_{\alpha=1}^{M} \left\{ \sum_{j=0}^{C} \frac{\nu_{\alpha j}}{T}(w_j - w_0)^2 - \left( \sum_{j=0}^{C} \frac{\nu_{\alpha j}}{T}(w_j - w_0) \right)^2 \right\} \right]^{1/2}$
    **return** $\mu$, $\sigma$
**end function**

---

where $\boldsymbol{w}$ is a $(C + 1)$-dimensional vector of constant weights. For example, if $w_k = k$, then $\bar{\pi}$ represents the average category label. In the case where $C = 1$, this average corresponds to the mean probability of a correct answer over the entire test set. However, we allow for a general choice of $\boldsymbol{w}$ to accommodate a wide range of possible metrics.

## 2.4 BAYESIAN ESTIMATOR AND UNCERTAINTY FOR THE PERFORMANCE METRIC

In principle, we could estimate $\pi_\alpha$ by running an arbitrarily large number of trials with the LLM, yielding an accurate estimate of $\bar{\pi}$. However, we are typically constrained to small $N$ due to limited computational resources. Our goal is to develop a Bayesian approach to estimate $\bar{\pi}$ and its associated uncertainty given a finite $N$. The first step is to construct $\mathcal{P}(\boldsymbol{\pi}_\alpha | \boldsymbol{R}_\alpha)$, the posterior probability of $\boldsymbol{\pi}_\alpha$ given the $\alpha$th row of the matrix $R$, denoted $\boldsymbol{R}_\alpha$. This posterior depends on the data in $\boldsymbol{R}_\alpha$ and a chosen prior distribution $\mathcal{P}(\boldsymbol{\pi}_\alpha)$ for the unknown underlying probability vector $\boldsymbol{\pi}_\alpha$. The prior could be uniform (assuming no prior information) or incorporate previously gathered evidence about the LLM's performance. The Bayesian framework focuses on two quantities: the first is the mean of $\bar{\pi}$ over the joint posterior for all questions, which we denote as $\mu(R)$. This is a Bayesian optimal estimator, minimizing the quadratic loss function $\mathcal{L}(\bar{\pi}^{\text{est}}) = \mathbb{E}_{R, \boldsymbol{\pi}_\alpha}(\bar{\pi}^{\text{est}}(R) - \bar{\pi})^2$ over all possible estimators $\bar{\pi}^{\text{est}}(R)$, where the expectation value is over all possible $\boldsymbol{\pi}_\alpha$ and realizations of $R$ (41). The second quantity is the variance $\sigma^2(R)$, which quantifies the uncertainty of the $\mu$ estimate. Both $\mu(R)$ and $\sigma^2(R)$ have exact closed-form expressions, derived in Appendix A, and can be simply calculated for any $R$ using Algorithm 1.

## 2.5 USING UNCERTAINTY ESTIMATES TO DECIDE SIGNIFICANCE OF PERFORMANCE DIFFERENCES

In general, the expressions for $\mu(R)$ and $\sigma^2(R)$ are valid for any $M$ and $N$, and do not rely on asymptotic arguments like the central limit theorem (CLT). However, there are useful simplifications that occur in specific limiting cases. For example, as the size of the test set $M$ becomes large, we can derive not just the moments of the posterior distribution for $\bar{\pi}$, but also its shape, which becomes approximately Gaussian: $\mathcal{P}(\bar{\pi}|R) \sim \mathcal{N}(\mu(R), \sigma^2(R))$. This allows us to assess whether two methods exhibit a statistically significant performance difference. Consider results matrices $R$ and $R'$ from two approaches, with corresponding means $\mu$, $\mu'$ and standard deviations $\sigma$, $\sigma'$. The distribution of the performance difference $\Delta \bar{\pi} \equiv \bar{\pi} - \bar{\pi}'$ is a convolution of the individual posteriors, yielding another normal distribution: $\mathcal{P}(\Delta \bar{\pi} | R, R') \sim \mathcal{N}(\tilde{\mu}, \tilde{\sigma}^2)$, where the mean of the difference is $\tilde{\mu} = \mu - \mu'$, and the standard deviation is $\tilde{\sigma} = \sqrt{\sigma^2 + (\sigma')^2}$. To determine our

confidence in the ranking of the two methods, we need to determine the probability that $\text{sign}(\Delta\bar{\pi}) = \text{sign}(\mu - \mu')$. This can be done by calculating the absolute $z$-score, $z = |\mu - \mu'|/\sqrt{\sigma^2 + (\sigma')^2}$. The probability that the ranking based on $\mu$ and $\mu'$ is correct (the ranking confidence $\rho$) is given by $\rho = (1/2)(1 + \text{erf}(z/\sqrt{2}))$. For example $z = 1.645$ corresponds to $\rho = 0.95$.

## 2.6 EQUIVALENCE OF BAYESIAN AND AVERAGE RANKINGS FOR UNIFORM PRIOR

In the results below, we will denote ranking based on the Bayesian estimator $\mu$ with a uniform prior as Bayes@$N$. Because $\mu$ is related to a naive weighted average accuracy via a positive affine transformation, it turns out the ranking based on the average, denoted as avg@$N$, is identical to Bayes@$N$ (for the detailed proof, see Appendix B). In the large-trial limit $N \to \infty$, the value of $\mu$ approaches the average, as expected, but the ranking equivalence holds at all finite $N$. This relationship also extends to uncertainty quantification, where the standard deviation of the average relates to the Bayesian standard deviation $\sigma$ by a scaling factor, providing a concrete method to compute uncertainty in the average without relying on the Central Limit Theorem. This is particularly advantageous in small-sample regimes common in LLM evaluations, where CLT-based methods often underestimate uncertainty and produce invalid intervals (e.g., extending beyond [0,1] or collapsing to zero) (42). As highlighted by (42), Bayesian approaches with uniform priors (e.g., Beta(1,1) in the binary case) yield well-calibrated credible intervals even for datasets with fewer than a few hundred datapoints, outperforming CLT approximations in coverage and handling complex structures like clustered data.

## 2.7 GOLD STANDARD FOR RANKING

Strictly speaking, the underlying true ranking of LLMs for a particular performance metric $\bar{\pi}$ is unknown, because it would require determining the infinite trial limit, $\bar{\pi} = \lim_{N \to \infty} \mu$, for each LLM. In practice, we have to settle for an approximation to $\bar{\pi}$, calculated at some large but finite value $N = N_{\max}$ (for example $N_{\max} = 80$ in our LLM experiments). Specifically, we use Bayes@$N_{\max}$—which is the same as the ranking based on avg@$N_{\max}$—as our "gold standard" or reference ranking (43). In other words, rankings using smaller $N$ will be compared to this gold standard to assess their accuracy.

For this comparison, we employ Kendall's $\tau$, a nonparametric correlation coefficient that measures ordinal agreement between two rankings by comparing the number of concordant and discordant pairs of models. The coefficient ranges from $-1$ (perfect inversion) to $+1$ (perfect agreement), with 0 indicating no association. We specifically use the $\tau_b$ variant, which properly accounts for ties in the rankings (e.g., the intentional tie in our simulation below), ensuring that equivalences do not artificially inflate the correlation. See Appendix I.1 for further discussion and formal definitions.

To validate our claims about Bayes@$N_{\max}$ as a gold standard and determine which methods converge to the true ranking, we run a biased-coin simulation in which the success probabilities $\boldsymbol{\pi}_\alpha$ and mean performance $\bar{\pi}$ are known. We generate 11 sets of 30 probabilities, with $\bar{\pi}$ values of $[0.2332, 0.2545, 0.3604, 0.3642, 0.3642, 0.4466, 0.5418, 0.5276, 0.608, 0.6213, 0.7327]$, representing different LLMs and including a tie at $0.3642$ to test equivalent performances. For $M = 30$ questions and up to 80 trials, panel (a) computes Pass@$k$ ($k = 2, 4, 8$), Bayes@$N$, Pass^$k$, G-Pass@$k_{\bar{\tau}}$, and mG-Pass@$k$ scores over 1000 independent $R$ matrices, derives rankings, and reports their average Kendall's $\tau$ against the gold standard. We omit avg@$N$ because it is rank-equivalent to Bayes@$N$ (Section 2.6). In practice, we are limited to a small number of trials per question. To study the $N = 80$ regime, we bootstrap a single $R$ matrix with replacement and compare the resulting convergence curves to the ideal baseline from many independent $R$ matrices (panel a). For each $N = 1, \ldots, 80$, we generate 10,000 replicates under two schemes: column-wise bootstrapping, which resamples trial indices (panel b), and row-wise bootstrapping, which resamples answers independently for each question (panel c). For each replicate, we recompute scores, rankings, and $\tau$, then average them to obtain smoothed curves. The two schemes give nearly identical results and closely match panel (a), showing that $\tau$ convergence is insensitive to answer ordering in the rows or columns of $R$. Although unnecessary in our LLM mimic simulations, bootstrapping provides a practical way to estimate convergence from limited real LLM trial data.

As seen in Figure 2, Bayes@$N$ starts with relatively high agreement with the gold standard and reaches $\tau = 1$ much faster than Pass@$k$ and its variants, which have greater variance and bias at

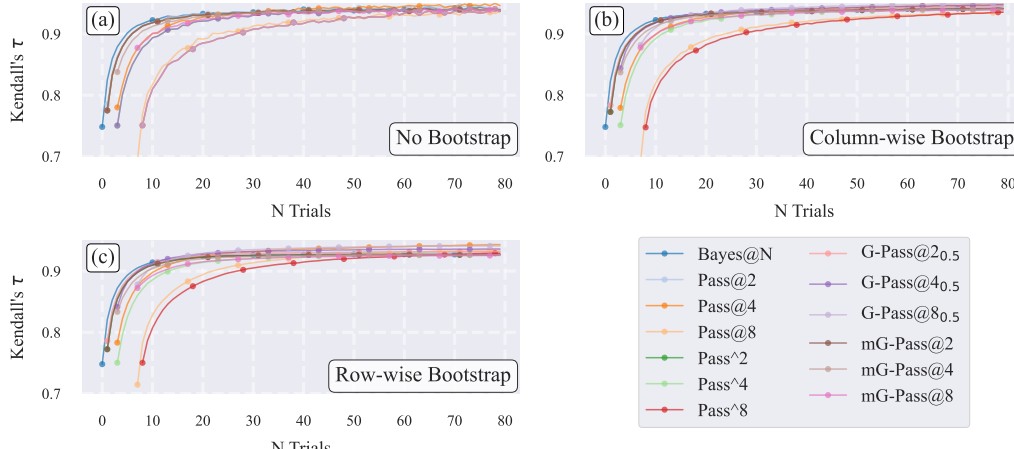

Figure 2: Kendall's $\tau$ rank correlation for various evaluation methods compared to the true ranking of 11 sets of biased coins (LLM mimics) with known mean success probabilities $\bar{\pi} = 0.2332$, $0.2545$, $0.3604$, $0.3642$, $0.3642$, $0.4466$, $0.5418$, $0.5276$, $0.608$, $0.6213$, $0.7327$. The simulation evaluates methods including Pass@$k$ ($k = 2, 4, 8$), Bayes@$N$, Pass^$k$, G-Pass@$k_{\tilde{\tau}}$ ($\tilde{\tau} = 0.5$), and mG-Pass@$k$ across 1 to 80 trials. Panel a) shows $\tau$ results without bootstrapping, while panels b) and c) use two different bootstrapping approaches with $10^4$ samples.

small $N$. Although all methods eventually recover the same ranking, their convergence rates differ substantially, making sample efficiency central to choosing an evaluation method. Here we use uniform priors, but faster convergence may be possible with non-uniform priors informed by correlated models (e.g., base, older, or quantized versions); Appendix C gives a preliminary synthetic demonstration.

### 2.7.1 RANKING WITH UNCERTAINTY

In Section 2.5, we described how uncertainty estimates from the Bayesian approach can be used to evaluate the relative performance of two models. Here, we extend these ideas to incorporate uncertainty into the ranking of multiple models. We do this via our biased-coin LLM mimics, which we denote $\text{LLM}_\beta$ for $\beta = 1, \ldots, 11$, described in the previous section. To incorporate a chosen credible interval in the ranking, we order their $\mu$ values from highest to lowest, choose the appropriate $z$ threshold (for example $z = 1.645$ for 95% CI in the ranking), and assign two consecutive methods the same ranking if the absolute $z$-score falls below this threshold.

The first row of Table 1 shows the underlying gold standard ranking for all the LLM mimics, since in this case we know the true $\bar{\pi}$ values. Note the tie between $\text{LLM}_4$ and $\text{LLM}_5$, because their $\bar{\pi} = 0.3642$ is the same. The second row shows the Bayes@80 ranking without a credible interval (CI) and the third row shows Bayes@80 incorporating the 95% CI. The Bayes@80 ranking without CI aligns with the gold standard, except for two differences: the order of $\text{LLM}_{10}$ and $\text{LLM}_9$ is swapped, and the tie between $\text{LLM}_5$ and $\text{LLM}_4$ is not captured, which is expected since this ranking relies solely on $\mu$ estimates without accounting for uncertainty $\sigma$. In contrast, the third row, which incorporates the CI, reveals multiple ties across several models. Interestingly, $\text{LLM}_{10}$ and $\text{LLM}_9$ are now indistinguishable at the 95% CI. Despite the fact that $N = 80$ would be an atypically large number of trials for an actual LLM evaluation, it is insufficient to confidently distinguish the small performance difference ($\bar{\pi} = 0.608$ vs. $0.6213$) between the two models. In Appendix D we show that it would actually require increasing $N$ by a factor of 3 to achieve 95% CI, highlighting the difficulties of reliably ranking models with similar performance.

## 3 EXPERIMENTS

In this section, we empirically validate our proposed evaluation methods using real-world datasets, focusing on ranking LLMs for mathematical reasoning tasks. We employ bootstrapping to com-

Table 1: Comparison of biased-coin LLM mimic rankings based on the gold standard, Bayes@80 without credible interval (CI), and Bayes@80 with CI.

| LLM mimic | $LLM_{11}$ | $LLM_{10}$ | $LLM_9$ | $LLM_8$ | $LLM_7$ | $LLM_6$ | $LLM_5$ | $LLM_4$ | $LLM_3$ | $LLM_2$ | $LLM_1$ |
|---|---|---|---|---|---|---|---|---|---|---|---|
| **Gold Standard** | 1 | 2 | 3 | 5 | 4 | 6 | 7 | 7 | 8 | 9 | 10 |
| **Bayes@80 (w/o CI)** | 1 | 3 | 2 | 5 | 4 | 6 | 7 | 8 | 9 | 10 | 11 |
| **Bayes@80 (w/ CI)** | 1 | 2 | 2 | 3 | 3 | 4 | 5 | 5 | 5 | 6 | 7 |

pute the expected value of each evaluation score at a given $N$. First, we present rankings of LLMs on the AIME'24, AIME'25, BrUMO'25, and HMMT'25 datasets without accounting for variance, based solely on evaluation scores (with ties occurring when scores are identical). Subsequently, we demonstrate how incorporating uncertainty in these scores can alter rankings across different datasets. Building on the discussion in Section 2.7, we adopt the ranking derived from avg@80 (equivalently, Pass@1 evaluated on the same 80 trials) or Bayes@80 (uniform-prior Bayesian estimator) as our gold standard for comparing current LLMs, noting their equivalence in rankings (as proven in Section 2.6). For each $N$ from 1 to 80 (with Pass@k and similar methods starting from $N = k$ to avoid computation with insufficient samples), we compare the rankings produced by various evaluation methods against this gold standard, reporting the average Kendall's $\tau$ over $10^4$ bootstrapped resamples to estimate the expected rank correlation at each step (assuming independence among questions and trials).

## 3.1 CONVERGENCE TO GOLD STANDARD

To assess the ability of different evaluation methods to compare the performance of different LLMs, we plot the average Kendall's $\tau$ against the gold standard as a function of the number of trials $N$ in Figure 3, combining results from AIME'25 (panel a), AIME'24 (panel b), HMMT'25 (panel c), and BrUMO'25 (panel d). Across all datasets, the Bayes@$N$ and avg@$N$ curves overlap completely (so we only plot Bayes@$N$) and demonstrate the fastest convergence to high $\tau$ values, indicating robust alignment with the gold standard even in low-sample regimes. In all four datasets, Bayes@$N$ reaches $\tau > 0.90$ by $N = 10$ and approaches $\tau \approx 1$ at $N \approx 80$. The only exception is AIME'25, where $\tau > 0.90$ is achieved by $N = 10$, but the curve converges to $\tau \approx 0.95$ at $N = 80$.

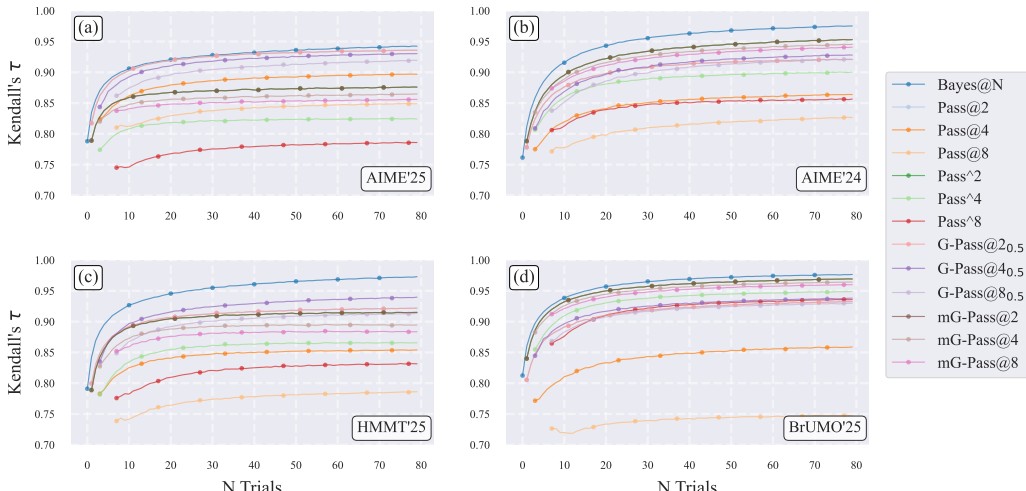

Figure 3: Average Kendall's $\tau$ correlation between rankings produced by various evaluation methods and the gold standard (derived from Bayes@80, or equivalently avg@80), as a function of the number of trials $N$. Results are averaged over $10^4$ bootstrapped resamples for each dataset: (a) AIME'25, (b) AIME'24, (c) HMMT'25, and (d) BrUMO'25. Methods include Bayesian estimation Bayes@$N$, Pass@k ($k = 2, 4, 8$), Pass^k, G-Pass@$k_{\tilde{\tau}}$ ($\tilde{\tau} = 0.5$), and mG-Pass@k.

In contrast, Pass@$k$ variants ($k = 2, 4, 8$) and their variations (e.g., Pass^$k$, G-Pass@$k_{\tilde{\tau}}$ with $\tilde{\tau} = 0.5$, mG-Pass@$k$) start with lower Kendall's $\tau$ compared to Bayes@$N$ and converge more slowly in all four datasets. At every $N$, Bayes@$N$ consistently shows faster convergence and higher agreement with the gold standard. These findings align with our biased-coin simulations in Section 2.7, demonstrating that the Bayesian method best satisfies the gold-standard criteria—low uncertainty, minimal ties, and rapid convergence—across diverse mathematical reasoning benchmarks.

## 3.2 RANKINGS WITH CREDIBLE INTERVALS

Following the methodology of Section 2.7.1, we compare model rankings across four datasets (AIME'25, AIME'24, HMMT'25, and BrUMO'25) using Bayes@80 as the gold standard (see Figure 3). Table 2 summarizes these comparisons by reporting, for each dataset, two versions of the ranking: the rank *with* a 95% CI and the rank *without* CI. The "w/ CI" rank accounts for uncertainty in the Bayes@80 scores and therefore allows models with overlapping CIs to share the same rank; the "w/o CI" rank is the strict ordering determined by the point estimates of Bayes@80 for that dataset.

Table 2 indicates that point-estimate rankings diverge from those accounting for credible intervals. Qwen3-30B-A3B-Thinking-2507 and Qwen3-4B-Thinking-2507 consistently secure the top positions across all four datasets; specifically, the dominance of the 30B model is statistically distinguishable at the 95% CI level in every case. Conversely, the relative ordering of the remaining models varies by dataset.

When incorporating 95% CIs, we observe that while all four datasets exhibit five tied groups, the extent of ambiguity varies significantly. AIME'25 yields the fewest distinct ranks (up to 11), followed by AIME'24 (up to 13), and both HMMT'25 and BrUMO'25 (up to 14). This compression of ranks indicates greater uncertainty in the Bayes@80 gold standard for AIME'25 (due to more extensive ties) compared to the others under our current trial budget. Intuitively, this higher uncertainty in AIME'25's gold-standard scores implies that more additional trials would be required for that dataset to empirically produce a statistically stable ranking; conversely, we can be more confident in the estimated gold standards for AIME'24, HMMT'25, and BrUMO'25 given the current number of trials. This distinction also explains why AIME'25 reaches a Kendall's $\tau$ of 0.95 at $N = 80$, whereas the other three datasets converge to $\sim 1$ at the same sample size in Figure 3.

Table 2: Rankings for four datasets. Models are listed in the order of their gold-standard ranking (Bayes@80 point estimates, i.e., without uncertainty) for AIME'25. Each dataset column gives the rank with a 95% CI (left) and the rank without CI (right).

| Model | AIME'25 | | AIME'24 | | HMMT'25 | | BrUMO'25 | |
|---|---|---|---|---|---|---|---|---|
| | w/ CI | w/o CI | w/ CI | w/o CI | w/ CI | w/o CI | w/ CI | w/o CI |
| Qwen3-30B-A3B-Thinking-2507 | 1 | 1 | 1 | 1 | 1 | 1 | 1 | 1 |
| Qwen3-4B-Thinking-2507 | 2 | 2 | 2 | 2 | 2 | 2 | 2 | 2 |
| gpt-oss-20b-high | 3 | 3 | 3 | 5 | 3 | 4 | 6 | 11 |
| gpt-oss-20b-medium | 3 | 4 | 3 | 3 | 2 | 3 | 7 | 12 |
| Phi-4-reasoning-plus | 3 | 5 | 3 | 4 | 3 | 5 | 3 | 5 |
| AceReason-Nemotron-1.1-7B | 4 | 6 | 5 | 9 | 4 | 6 | 3 | 4 |
| Phi-4-reasoning | 5 | 7 | 5 | 10 | 5 | 8 | 4 | 7 |
| gpt-oss-20b-low | 5 | 8 | 6 | 12 | 11 | 17 | 11 | 17 |
| OpenThinker2-32B | 5 | 9 | 4 | 8 | 5 | 7 | 2 | 3 |
| Light-R1-14B-DS | 5 | 10 | 4 | 6 | 6 | 11 | 4 | 8 |
| FuseO1-DeepSeekR1-QwQ-SkyT1-Flash-32B | 5 | 11 | 4 | 7 | 6 | 9 | 3 | 6 |
| NVIDIA-Nemotron-Nano-9B-v2 | 6 | 12 | 6 | 11 | 6 | 10 | 5 | 10 |
| LIMO-v2 | 6 | 13 | 7 | 13 | 7 | 12 | 5 | 9 |
| EXAONE-4.0-1.2B | 7 | 14 | 8 | 14 | 7 | 13 | 10 | 15 |
| OpenR1-Distill-7B | 7 | 15 | 9 | 15 | 10 | 16 | 8 | 13 |
| OpenThinker3-1.5B | 8 | 16 | 10 | 16 | 8 | 14 | 9 | 14 |
| OpenReasoning-Nemotron-1.5B | 8 | 17 | 11 | 17 | 9 | 15 | 10 | 16 |
| DeepSeek-R1-Distill-Qwen-1.5B | 9 | 18 | 12 | 19 | 12 | 18 | 13 | 19 |
| Sky-T1-32B-Flash | 10 | 19 | 12 | 18 | 13 | 19 | 12 | 18 |
| Bespoke-Stratos-7B | 11 | 20 | 13 | 20 | 14 | 20 | 14 | 20 |

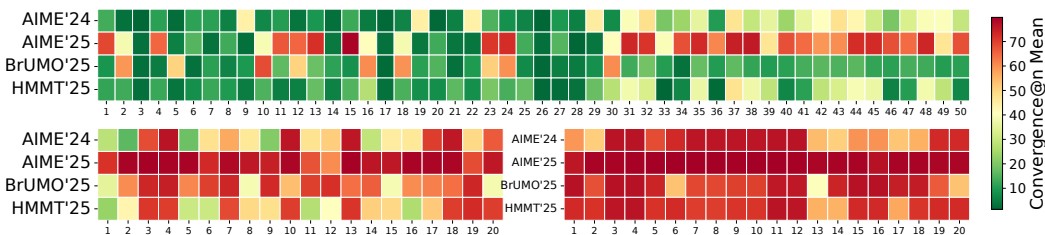

Figure 4: **Convergence@$n$ without CI**. Mean convergence@$n$ across model combinations for AIME'24, AIME'25, HMMT'25, and BrUMO'25. **Top**: 50 combinations of 5 models. **Bottom-left**: 20 combinations of 10 models. **Bottom-right**: 20 combinations of 15 models. Color indicates the mean convergence@$n$ over $10^5$ bootstrap replicates (green: fast convergence; red: slow convergence).

## 3.3 CONVERGENCE

In this section, we investigate model-ranking convergence, building on the showcase figure (Figure 1). We define convergence@$n$ as the smallest trial $n$ at which the ranking induced by the first $n$ trials matches the 80-trial *gold standard* and remains unchanged thereafter; lower values therefore indicate more sample-efficient ranking.

Lower convergence@$n$ values indicate that fewer trials are sufficient to achieve stable rankings. As detailed in the caption of Figure 1, the figure displays the probability mass functions (PMFs) of convergence@$n$ for each method across the datasets. These PMFs are empirically estimated by generating $10^5$ column-wise bootstrap replicates through resampling the $N_{\max}$ trials, then for each replicate, cumulatively evaluating the ranking at every $N$ (from 1 to 80) and identifying the minimal $n$ where the ranking stabilizes to the *gold standard*. This process captures the distribution of convergence points under repeated sampling, reflecting the inherent uncertainty in finite-sample rankings due to stochastic trial outcomes.

This bootstrapping approach provides a distribution over possible convergence points ($n$), offering insights into the variability and reliability of each evaluation method: Pass@$k$ (for $k = 2, 4, 8$) versus our Bayes@$N$. A lower mean convergence@$n$ signifies more cost-effective convergence, while failure to converge within 80 trials (as seen in AIME'25) indicates more trials are needed to *confidently* rank LLMs or we must include CI for a reliable ranking. The key result is that Bayes@$N$ converges reliably on all datasets except AIME'25, often with fewer trials than Pass@$k$. On HMMT'25 and BrUMO'25, for example, Bayes@$N$ reaches mean convergence at approximately $44.2$ and $27.1$ trials, compared to around $69.5$ and $48.5$ for the best-performing Pass@$k$ scores. Failure to converge within 80 trials, as in AIME'25, indicates that additional trials or CI-aware ranking are needed. Appendix J (Figure 9) gives the corresponding cumulative distribution functions (CDFs).

**Worst-case scenarios** To further distinguish the Bayes@$N$ framework from avg@$N$, we analyze the *worst-case* bootstrap replicates, i.e., those that either require the maximum number of trials to stabilize or fail to converge. For 11 LLMs, Figure 10 shows competition-ranking trajectories as trials accumulate. In AIME'24 (panel a) the ranking converges at trial 75, in BrUMO'25 (panel c) at trial 68, and in HMMT'25 (panel d) at trial 78, whereas in AIME'25 (panel b) no convergence is observed within 80 trials. This persistent instability motivates either additional trials or Bayes@$N$'s credible intervals to quantify uncertainty and estimate the minimum reliable $N$ (see Section 2.7.1).

This situation becomes even more severe as more models are included. As shown in Figure 11, when the number of models is increased to $L = 20$, none of the datasets exhibit convergence. To examine convergence as a function of $L$ more systematically, we consider a pool of 20 LLMs (Table 7) and construct 50 subsets of 5 models (Table 9), 20 subsets of 10 models (Table 10), and 20 subsets of 15 models (Table 11). For each subset, we generate $10^5$ bootstrap replicates to estimate convergence@$n$. Figure 4 reports the resulting convergence@$n$ values across all subsets and replicates, showing that as the number of models increases, evaluation methods such as avg@$N$ and the Pass@$k$ family become unreliable for estimating model abilities and producing stable rankings.

Table 3: Comparison of the Bayesian framework and other evaluation methods.

| Methods ($N$ trials) | Convergence | Credible interval | Prior knowledge | Categorical |
|---|---|---|---|---|
| Pass@$k$ and alternatives | ✗ | ✗ | ✗ | ✗ |
| avg@$N$ | ✓ | Limited | ✗ | ✗ |
| Bayes@$N$ | ✓ (Sec. 3.3, Figs. 1 and 4) | ✓ (Fig. 6, Table 1,2) | ✓(App. C) | ✓ (Sec. 3.4) |

### 3.4 RUBRIC-AWARE CATEGORICAL EVALUATION

While evaluation is often reduced to binary correctness, this simplification discards valuable signals that capture other aspects of model behavior. For instance, LLM outputs can be assessed not only on correctness but also on whether they are well-structured, coherent, or exhibit step-by-step reasoning in mathematical tasks. In practice, evaluators routinely record richer dimensions such as format compliance, calibration of confidence, degenerate outputs, out-of-distribution (OOD) behavior, and verifier scores. This limitation is especially important for reasoning models, where overthinking (44) inflates token usage without corresponding gains in reliability. Bayes@$N$ provides a principled way to capture these richer outcomes. By treating per-item results as categorical rather than binary, the approach aligns more closely with actual goals while preserving statistical rigor and transparency. This method enables a nuanced understanding of model performance across diverse dimensions, offering insights into trade-offs between correctness, efficiency, and robustness. For a comprehensive discussion of the categorical Bayesian evaluation framework, including base signals, schema definitions, and their impact on model rankings, see Appendix F.

## 4 RELATED WORK

Functional-correctness evaluation with Pass@$k$ became standard in code generation after HumanEval, where a task is counted as solved if any of $k$ sampled programs passes its tests (24). The same multi-sample evaluation style is now widely used for math reasoning and other verifiable LLM tasks (45; 46; 47; 27), and has motivated stricter variants such as $pass^\wedge k$, G-Pass@$k_\tau$, and mG-Pass (48; 27). These metrics capture useful notions of potential and consistency, but their estimates can remain unstable when the trial budget is small relative to $k$. Appendix H provides a fuller review of these metrics and their extensions.

Recent evaluation work also emphasizes reproducibility and uncertainty. HELM and the LM Evaluation Harness provide infrastructure for standardized, transparent benchmarking (5; 49), while interval-aware evaluation studies argue against over-interpreting point estimates on small benchmarks (42; 34; 50; 51). Our contribution is complementary: we give a closed-form Bayesian estimator and uncertainty estimate for binary and categorical outcomes, enabling ranking decisions that explicitly account for finite-sample uncertainty.

## 5 CONCLUSION: STRENGTHS, LIMITATIONS & FUTURE DIRECTIONS

The overall benefits of the Bayesian framework are summarized in Table 3: it provides fast convergence, analytical uncertainty estimates, and the incorporation of prior knowledge and categorical. Although we have validated our approach with biased-coin LLM mimic simulations, together with experiments using actual LLMs (up to $N_{\max} = 80$ trials across four tasks and 20 models), more extensive evaluations may be constrained by computing and academic budgets. The focus of the current work was the simplest version of the Bayesian approach, using a uniform prior, which provides a conservative and reproducible starting point. But the theory allows for more complex, informative priors, and this opens up a rich vein of future directions that should be systematically explored: for example priors from past runs, domain- or task-conditioned priors, and expert-elicited priors. These have the potential of accelerating convergence even further, but must be chosen and reported carefully. Clear guidance and tools for prior elicitation will hopefully ensure that gains in sample efficiency do not come at the cost of hidden bias.

ETHICS STATEMENT

This research relies only on publicly available, non-personal benchmarks; no human subjects, user data, or PII are involved. Potential misuse includes cherry-picking priors, rubrics, or samples to exaggerate performance. To prevent this, use of Bayes@$N$ with user-defined priors requires clear documentation and reporting of posterior credible intervals.

REPRODUCIBILITY STATEMENT

To ensure reproducibility, detailed implementation instructions are provided in Appendix I.

ACKNOWLEDGMENTS

This research was supported in part by NSF awards 2117439, 2112606, and 2320952.

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

CONTENTS

## A   DERIVATION OF BAYESIAN ESTIMATOR AND UNCERTAINTY

As described in the main text, the Bayesian framework is built on two quantities. The first is $\mu(R)$, the average of $\bar{\pi}$ over the joint posterior for all the questions:

$$\mu(R) = \int_{\Delta} d\boldsymbol{\pi}_1 \cdots \int_{\Delta} d\boldsymbol{\pi}_M \, \bar{\pi} \prod_{\alpha=1}^{M} \mathcal{P}(\boldsymbol{\pi}_\alpha | \boldsymbol{R}_\alpha), \tag{2}$$

where the integration region $\Delta$ is the probability simplex defined as the set of all possible $(C+1)$-dimensional vectors $\boldsymbol{p}$ such that $\sum_{k=0}^{C} p_k = 1$. The second is the variance $\sigma^2(R)$ associated with our Bayesian estimator,

$$\sigma^2(R) = \int_{\Delta} d\boldsymbol{\pi}_1 \cdots \int_{\Delta} d\boldsymbol{\pi}_M \, (\bar{\pi} - \mu(R))^2 \prod_{\alpha=1}^{M} \mathcal{P}(\boldsymbol{\pi}_\alpha | \boldsymbol{R}_\alpha). \tag{3}$$

Our derivation of closed-form expressions for $\mu$ and $\sigma$ builds on the generalized $(C > 1)$ and original $(C = 1)$ Laplace rule of succession theory from (41), recovering those results in the special case of a single question $(M = 1)$. We start with Bayes' rule for each row of $R$:

$$\mathcal{P}(\boldsymbol{\pi}_\alpha | \boldsymbol{R}_\alpha) = \frac{\mathcal{P}(\boldsymbol{R}_\alpha | \boldsymbol{\pi}_\alpha) \mathcal{P}(\boldsymbol{\pi}_\alpha)}{\mathcal{P}(\boldsymbol{R}_\alpha)}. \tag{4}$$

The likelihood $\mathcal{P}(\boldsymbol{R}_\alpha | \boldsymbol{\pi}_\alpha)$ is a $(C+1)$-category multinomial distribution over $N$ trials, with the probability distribution function:

$$\mathcal{P}(\boldsymbol{R}_\alpha | \pi_\alpha) = \frac{N!}{n_{\alpha 0}! n_{\alpha 1}! \cdots n_{\alpha C}!} \prod_{k=0}^{C} (\pi_{\alpha k})^{n_{\alpha k}}, \tag{5}$$

where $n_{\alpha k} = \sum_{i=1}^{N} \delta_{k, R_{\alpha i}}$, $\boldsymbol{n}_\alpha$ is the vector with elements $n_{\alpha k}$, and $\delta_{i,j}$ is the Kronecker delta.

The prior $\mathcal{P}(\boldsymbol{\pi}_\alpha)$ is chosen as the conjugate prior of the multinomial, a Dirichlet distribution $\mathcal{P}(\boldsymbol{\pi}_\alpha) \sim \mathrm{Dir}(\boldsymbol{n}_\alpha^0)$, with concentration parameter vector $\boldsymbol{n}_\alpha^0 = (n_{\alpha 0}^0, \ldots, n_{\alpha C}^0)$. (35) A uniform prior (no prior knowledge) sets $n_{\alpha k}^0 = 1$ for all $k$. Prior information from an earlier $M \times D$ matrix $R^0$ (with $R_{\alpha i}^0$ as the category for the $i$th trial of the $\alpha$th question) can be incorporated as:

$$n_{\alpha k}^0 = 1 + \sum_{i=1}^{D} \delta_{k, R_{\alpha i}^0}. \tag{6}$$

The Dirichlet prior is:

$$\mathcal{P}(\boldsymbol{\pi}_\alpha) = \frac{\Gamma(1 + C + D)}{\prod_{k=0}^{C} \Gamma(n_{\alpha k}^0)} \prod_{k=0}^{C} (\pi_{\alpha k})^{n_{\alpha k}^0 - 1}, \tag{7}$$

where $\sum_{k=0}^{C} n_{\alpha k}^0 = 1 + C + D$.

The normalization constant $\mathcal{P}(\boldsymbol{R}_\alpha)$ is:

$$\mathcal{P}(\boldsymbol{R}_\alpha) = \int_{\Delta} d\boldsymbol{p} \, \mathcal{P}(\boldsymbol{R}_\alpha | \boldsymbol{p}) \mathcal{P}(\boldsymbol{p}), \tag{8}$$

and since the Dirichlet is the conjugate prior, the posterior is $\mathcal{P}(\boldsymbol{\pi}_\alpha | \boldsymbol{R}_\alpha) \sim \mathrm{Dir}(\boldsymbol{\nu}_\alpha)$, with $\boldsymbol{\nu}_\alpha = \boldsymbol{n}_\alpha + \boldsymbol{n}_\alpha^0$. The posterior distribution is:

$$\mathcal{P}(\boldsymbol{\pi}_\alpha | \boldsymbol{R}_\alpha) = \frac{\Gamma(T)}{\prod_{k=0}^{C} \Gamma(\nu_{\alpha k})} \prod_{k=0}^{C} (\pi_{\alpha k})^{\nu_{\alpha k} - 1}, \tag{9}$$

where $T \equiv \sum_{k=0}^{C} \nu_{\alpha k} = 1 + C + D + N$.

The moment generating function $\Phi(t) = \langle \exp(\bar{\pi}t) \rangle$ is:

$$\Phi(t) = \int_\Delta d\boldsymbol{\pi}_1 \cdots \int_\Delta d\boldsymbol{\pi}_M \exp(t\bar{\pi}) \prod_{\alpha=1}^M \mathcal{P}(\boldsymbol{\pi}_\alpha | \boldsymbol{R}_\alpha)$$

$$= \prod_{\alpha=1}^M \int_\Delta d\boldsymbol{\pi}_\alpha \exp\left(\frac{t}{M}\sum_{k=0}^C w_k \pi_{\alpha k}\right) \mathcal{P}(\boldsymbol{\pi}_\alpha | \boldsymbol{R}_\alpha) \tag{10}$$

$$= e^{tw_0} \prod_{\alpha=1}^M \int_\Delta d\boldsymbol{\pi}_\alpha \exp\left(t\sum_{k=1}^C s_k \pi_{\alpha k}\right) \mathcal{P}(\boldsymbol{\pi}_\alpha | \boldsymbol{R}_\alpha),$$

where $s_k \equiv (w_k - w_0)/M$, and $\pi_{\alpha 0} = 1 - \sum_{k=1}^C \pi_{\alpha k}$.

Each integral is the moment-generating function for a Dirichlet distribution, expressed via the confluent Lauricella hypergeometric function $\Psi^{[C]}$:

$$\Phi(t) = e^{tw_0} \prod_{\alpha=1}^M \Psi^{[C]}(\nu_{\alpha 1}, \ldots, \nu_{\alpha C}; T; ts_1, \ldots, ts_C), \tag{11}$$

where

$$\Psi^{[C]}(\nu_{\alpha 1}, \ldots, \nu_{\alpha C}; T; ts_1, \ldots, ts_C) = \sum_{m_1=0}^\infty \cdots \sum_{m_C=0}^\infty \frac{(\nu_{\alpha 1})_{m_1} \cdots (\nu_{\alpha C})_{m_C} (ts_1)^{m_1} \cdots (ts_C)^{m_C}}{(T)_m m_1! \cdots m_C!}, \tag{12}$$

and $(x)_n$ is the Pochhammer symbol.

The moments are:

$$\mu = \Phi'(0), \qquad \sigma^2 = \Phi''(0) - (\Phi'(0))^2. \tag{13}$$

Expanding $\Psi^{[C]}$ to $\mathcal{O}(t^2)$:

$$\Psi^{[C]} = 1 + \frac{t}{T}\sum_{j=1}^C \nu_{\alpha j} s_j + \frac{t^2}{2T(T+1)}\sum_{j=1}^C \nu_{\alpha j}(\nu_{\alpha j} + 1)s_j^2$$

$$+ \frac{t^2}{T(T+1)}\sum_{\ell=1}^C \sum_{m=\ell+1}^C \nu_{\alpha\ell}\nu_{\alpha m}s_\ell s_m + \mathcal{O}(t^3). \tag{14}$$

Substituting into equation 11 and computing derivatives yields:

$$\mu = w_0 + \frac{1}{MT}\sum_{\alpha=1}^M \sum_{j=0}^C \nu_{\alpha j}(w_j - w_0),$$

$$\sigma^2 = \frac{1}{M^2(T+1)}\sum_{\alpha=1}^M \left\{ \sum_{j=0}^C \frac{\nu_{\alpha j}}{T}(w_j - w_0)^2 - \left(\sum_{j=0}^C \frac{\nu_{\alpha j}}{T}(w_j - w_0)\right)^2 \right\}. \tag{15}$$

The algorithm summarizing this calculation is shown in Algorithm 1 in the main text.

## B   PROOF OF EQUIVALENCE OF BAYESIAN AND AVERAGE RANKINGS FOR UNIFORM PRIOR

For Bayesian estimators using a uniform prior (where $D = 0$, $T = 1 + C + N$, $\nu_{\alpha k} = 1 + n_{\alpha k}$), the expression for the mean $\mu$ from equation 15 simplifies as:

$$\mu = w_0 + \frac{1}{M(1+C+N)}\sum_{\alpha=1}^M \sum_{j=0}^C (1 + n_{\alpha j})(w_j - w_0)$$

$$= A + \frac{1}{M(1+C+N)}\sum_{\alpha=1}^M \sum_{j=0}^C w_j n_{\alpha j}, \tag{16}$$

where the constant $A$ is given by

$$A = \frac{1}{1+C+N} \sum_{j=0}^{C} w_j, \tag{17}$$

and $\sum_{j=0}^{C} n_{\alpha j} = N$. Here, $\mu$ relates to a naive weighted average accuracy $a$ over the number of answers in each category,

$$a = \frac{1}{MN} \sum_{\alpha=1}^{M} \sum_{j=0}^{C} w_j n_{\alpha j}, \tag{18}$$

via

$$\mu = A + \frac{N}{1+C+N} a. \tag{19}$$

Note that in the binary case where $C = 1$, $w_0 = 0$, $w_1 = 1$, the value of $a$ is just regular average accuracy avg@$N$. For categorical cases, it is just a weighted generalization of avg@$N$.

Since $A$ is constant across models and the prefactor $\frac{N}{1+C+N}$ is positive, we see that if $\mu > \mu'$, the corresponding values of $a$ and $a'$ from the two methods must always give the same ranking, $a > a'$. Additionally, in the limit of a large number of trials, $N \to \infty$, we see that $A \to 0$ and $\mu \approx a$, as expected.

This equivalence extends to uncertainty quantification. The relationship between the standard deviation of the average ($\sigma_{\text{avg@}N}$) and the Bayesian standard deviation ($\sigma_{\text{Bayes@}N}$ from equation 15) is

$$\sigma_{\text{avg@}N} = \frac{1+C+N}{N} \sigma_{\text{Bayes@}N}. \tag{20}$$

The Bayesian expression for $\sigma_{\text{Bayes@}N}$ is valid for all $M$ and $N$, providing a reliable method to compute uncertainty in avg@$N$ without relying on the Central Limit Theorem.

## C  POTENTIAL BENEFITS OF NON-UNIFORM PRIORS

While the convergence results in the main text demonstrate that Bayes@$N$ with a uniform prior outperforms alternatives like Pass@$k$ in ranking models, there are scenarios where non-uniform priors can achieve even faster convergence. This is the case when we have data from models that are related or closely correlated to the ones we are ultimately interested in ranking. Potential examples include: i) results from an older version of a model used as a prior for ranking a newer version; ii) a non-quantized version (where running trials is computationally expensive) used to provide prior data for a quantized version (where achieving large $N$ is cheaper); iii) a base model used to provide prior data for a fine-tuned one. Though a full exploration of these kinds of priors will be left to a future work, in this section we will show the potential benefits through our synthetic biased-coin LLM models, introduced in Section 2.7.

We start with a set of eight "original" models with $C = 1$, labeled by $i = 1, \ldots, 8$. Each model $i$ consists of a set of $M = 30$ success probabilities $\pi_{\alpha 1}$ drawn from a distribution Beta$(i + 3, 12 - i)$. We fix these probabilities for all the numerical experiments described below, and their averages for the eight models are: $\bar{\pi} = [0.3021, 0.3166, 0.4144, 0.4985, 0.5351, 0.5759, 0.6679, 0.7487]$. Hence for the original models higher $i$ corresponds to higher overall accuracy. We now imagine an "update" of model $i$ that mimics some kind of revision, fine-tuning, or other modification. Because the performance of the updated model should be correlated with the original, we model the update as a stochastic perturbation to the Beta distribution from which success probabilities are drawn: for updated model $i$ the $\pi_{\alpha 1}$ values are drawn from Beta$(i + 3 + \sigma, 12 - i + \sigma')$, where $\sigma = \pm 1$ and $\sigma' = \pm 1$ are random integers of unit magnitude. For the updated models the value of $\bar{\pi}$ may not strictly increase with $i$, so the ranking of models could be different than the original. Figure 5(a) shows a histogram of the Kendall's $\tau$ values comparing the original model set (described above) and 50k possible updated sets drawn using this stochastic procedure. A $\tau$ value of 1 corresponds to exactly the same ranking, and we see that the mean $\tau$ over the 50k realizations is 0.88. Hence

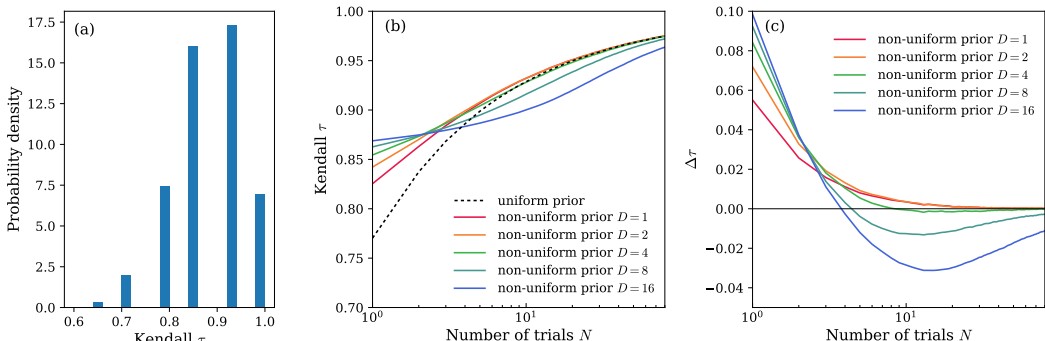

Figure 5: (a) Histogram of Kendall $\tau$ values comparing original ranking of synthetic LLM models and 50k replicates of updated models. (b) Mean Kendall $\tau$ between the estimated and true ranking for the updated models (50k replicates) as a function of $N$, the number of trials. The dashed line corresponds estimates using Bayes@$N$ with a uniform prior ($D = 0$), while the solid lines are Bayes@$N$ with a non-uniform prior and different choices of $D$. The non-uniform prior is based on results from $D$ trials of the original models. (c) Same as panel (b), except showing the difference $\Delta\tau$ between the non-uniform prior curves and the uniform curve.

there is some correlation between the original and updated rankings, but in the vast majority of cases (about 86% of the updates) the ranking has changed for the updated models.

The question we would like to ask is whether we can use the results from the original models as priors to help speed up convergence when ranking the updated models. To employ a non-uniform prior for a given model, we follow the procedure described in Appendix A, and incorporate the prior via the $M \times D$ results matrix $R^0$ corresponding to $D$ trial results over $M$ questions using the original model. Combined with $N$ trial results from the updated model, we get the Bayes@$N$ accuracy estimate $\mu$ for the updated model. These estimates are then used to rank the 8 updated models. Because we know the $\bar{\pi}$ values for this set, we know the true ranking, and we can compare the estimated and true rankings via Kendall's $\tau$.

For each choice of $N$ and $D$ we run 50k replicates, with each replicate consisting of a set of stochastic updates of the original models. The mean $\tau$ values over all these replicates are shown in Figure 5(b) as a function of $N$ for several different $D$. As expected, the $\tau$ curves increase with $N$, since the ranking becomes more certain with more trials, but the convergence properties vary. The dashed line is the case of a uniform prior ($D = 0$), while the solid lines represent five different non-uniform prior scenarios, with $D = 1$, 2, 4, 8, and 16. For small $N$ and small $D \leq 4$ we see a clear benefit of the non-uniform prior: already at $N = 1$ the value of $\tau$ starts higher than the uniform case, and remains so until the latter catches up for $N > 5$. Thus when we have prior data available, we can extract more accurate rankings with just a small number of trials of the updated model, relative to the uniform case. However there is a possibility to over-emphasize the prior: when $D = 8$ and 16, the benefit for small $N$ turns into a disadvantage at larger $N$. The $\tau$ curves dip beneath the $D = 0$ result, indicating that the prior has impeded accurate ranking. Figure 5(c) shows these trends more clearly by plotting $\Delta\tau$, the difference between the $\tau$ for each $D$ and the uniform $\tau$ with $D = 0$. So we see that priors have to be used judiciously, with large enough $D$ to nudge the ranking in the correct direction, but not too large to outweigh the results from the updated models. One of the goals of our future work will be to establish practical guidelines for $D$ in different real-world use cases.

## D   MODEL DISTINGUISHABILITY AND SAMPLE SIZE

To quantify the trials needed to reliably separate models with closely matched performance, we simulated the probability of correctly ranking $\text{LLM}_{10}$ above $\text{LLM}_9$ as a function of the number of trials $N$, shown in the left panel of Figure 6. At $N = 80$, the probability of obtaining the correct ranking is 83.7%. The right panel plots the absolute $z$-score versus $N$; at $N = 80$, $z \sim 1.14$, corresponding to approximately 87% CI (though the plots exhibit some noise due to simulation variability). These values closely align with the empirical probabilities in the left panels.

We also determined the minimum sample size $N$ needed to achieve z-scores of $1.645$ and $1.96$, corresponding to CI of approximately $95\%$ and $97.5\%$, respectively, for distinguishing between models. These thresholds occur at about $N = 199$ and $N = 285$. At these values, the simulated probability of correctly ranking the models is $94.7\%$ and $96.9\%$, respectively, which is closely consistent with expectations given the inherent noise in the results. These results underscore the computational cost of distinguishing models whose true performance metrics differ only slightly. In our biased-coin setup, the underlying success probabilities were $\bar{\pi}_9 = 0.608$ and $\bar{\pi}_{10} = 0.6213$, yet reliably establishing this distinction requires nearly 200 trials. Such large sample requirements highlight the importance of considering both uncertainty and convergence rates when interpreting ranking-based evaluations.

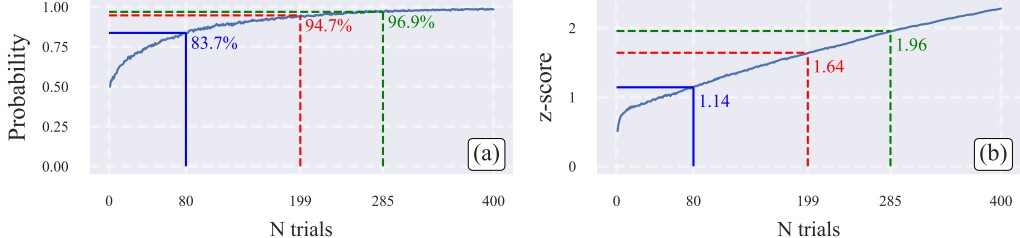

Figure 6: (a) Probability of correctly ranking $\mathrm{LLM}_{10}$ above $\mathrm{LLM}_9$ using Bayes@$N$ in the biased-coin simulations, shown as a function of trial count $N$. The probability is $83.7\%$ at $N = 80$, increases to $\sim 94.7\%$ at $N = 199$, and reaches $96.9\%$ at $N = 285$. (b) Corresponding absolute z-scores as a function of $N$, with values of $\sim 1.14$ at $N = 80$, $1.645$ at $N = 199$ ($95\%$ CI), and $1.96$ at $N = 285$ ($97.5\%$ CI).

## E   RUNTIME

To see the asymptotic runtime and memory scaling let:

$M$ = number of problems (rows),
$N$ = number of trials per problem (columns in $R$),
$D$ = number of prior outcomes per problem (columns in $R_0$, which may be $0$),
$C + 1$ = number of categories.

From Algorithm 1, the work is:

$$\text{Two row-wise histograms: } \mathcal{O}(MN) \text{ for } R \; + \; \mathcal{O}(MD) \text{ for } R_0,$$

$$\text{Posterior mean and variance on } \nu \in \mathbb{R}^{M \times (C+1)}: \quad \mathcal{O}(M(C+1)).$$

So the **overall time complexity** is:

$$\mathcal{O}(M(N + D + C))$$

i.e., linear in the number of entries in the result matrices and linear in the number of categories.

The **memory footprint** is likewise linear:

$$\text{Store } R \text{ and (optionally) } R_0: \quad \mathcal{O}(MN + MD),$$

$$\text{Store per-row category counts and derived arrays } (\nu, \nu/T): \quad \mathcal{O}(M(C+1)).$$

Note that the evaluation consists of tallying counts and then plugging them into closed-form expressions for $\mu$ and $\sigma$; no iterative optimization or Monte Carlo sampling is required.

# F CATEGORICAL EVALUATION

## F.1 RUBRIC-AWARE BAYES@N EVALUATION OF REASONING MODELS

As discussed in Section 2.3 and Section 3.4, for each question $\alpha \in \{1, \ldots, M\}$, every attempt yields base signals such as `has_box`, `is_correct`, `token_ratio`, `prompt_bpt`, `completion_bpt`, and verifier probabilities $compass\_context\_A, compass\_context\_B, compass\_context\_C$ for *correct*, *wrong*, and *invalid/off-task*. Using thresholds and Boolean criteria, each attempt is mapped into one of $C + 1$ categories under a chosen schema (e.g., *Format Aware*, *Conf-Wrong Penalty*, *Efficiency-Adjusted*; Table 5). We instantiate categorical schemata and update posterior means via Dirichlet–multinomial inference, yielding metrics that preserve correctness while explicitly reflecting formatting, calibration, and efficiency.

**Base signals**    All signals are directly obtainable from common LLM inference stacks such as Hugging Face transformers (52) and vLLM (16), via per-step scores/log-probs and termination metadata, and require no model-specific instrumentation; the verifier probabilities compass_context_A, compass_context_B, compass_context_C are defined in Appendix F.1.

- **has_box**: 1 if a final boxed answer is present; else 0.
- **is_correct**: 1 if the answer is correct; else 0.
- **token_ratio**: completion tokens normalized by 32,768.
- **repeated_pattern**: 0 if `finish_reason` is `stop`; else 1 (degenerate output).
- **prompt_bpt**: negative average prompt log-prob in bits/token (lower is better).
- **completion_bpt**: negative average completion log-prob in bits/token (lower is better).
- **compass_context_A**: verifier contextual probability of *correct*.
- **compass_context_B**: verifier contextual probability of *wrong*.
- **compass_context_C**: verifier contextual probability of *irrelevant/off-task*.

**Reward models in evaluation.**    While reward models are most familiar from fine-tuning (e.g., RLHF), we use one as a lightweight verifier to supply per-attempt label probabilities for

$$\{compass\_context\_A, compass\_context\_B, compass\_context\_C\}$$
$$= \{\text{correct, wrong, invalid/off-task}\}.$$

in evaluation. Concretely, we employ OpenCompass `CompassVerifier-3B` to produce probabilities and then apply *contextual calibration* to obtain a more robust, prompt-stable label distribution: we evaluate next-token scores for the candidate labels at a fixed answer slot, subtract a content-free baseline logit $b_y$ from the task logit $s_y$ for each label $y$, and apply temperature scaling to yield calibrated probabilities

$$p(y \mid x) = \text{softmax}\left(\frac{s_y - b_y}{T}\right).$$

This helps us mitigate saturation and the entanglement of formatting and confidence seen with last-token probabilities, and improves probability calibration for downstream rubric scoring.

**Selected categorical schema.**    We define 12 schemata (Table 5) using the rubric variables (Table 4) derived from the base signals; here are two illustrative definitions (the others follow analogously):

- **Format Aware**:
$$\text{cat} = \begin{cases} 0 & \text{invalid} \\ 1 & \text{wrong} \wedge \text{unboxed} \\ 2 & \text{wrong} \wedge \text{boxed} \\ 3 & \text{correct} \wedge \text{unboxed} \\ 4 & \text{correct} \wedge \text{boxed} \end{cases}$$

Table 4: Rubric variables, decision formulas, and brief descriptions used to map each model attempt into discrete categories. Thresholds ($\tau_{\text{high}}, \tau_{\text{low\_wrong}}, \tau_{\text{prompt}}$) and length quantiles ($\text{len\_p33}, \text{len\_p66}$) are computed per dataset from observed bits-per-token and token-ratio statistics. Category 0 is reserved for invalid outputs (degenerate repetition or high verifier $compass\_context\_C$), and $compass\_context\_A, compass\_context\_B, compass\_context\_C$ denote calibrated verifier probabilities for *correct*, *wrong*, and *off-task*, respectively.

| Rubric variables | Formula | Description |
|---|---|---|
| invalid | $(\text{repeated\_pattern} = 1) \vee (compass\_context\_C \geq 0.5)$ | Category 0 reserved for invalid. |
| correct | $(\text{is\_correct} \geq 0.5)$ | Boolean mask of correctness. |
| wrong | $(\text{is\_correct} < 0.5)$ | Complement of correct. |
| high_conf | $(\text{completion\_bpt} \leq \tau_{\text{high}})$ | Confidence proxy |
| low_conf | $(\text{completion\_bpt} > \tau_{\text{high}})$ | Complement of high_conf. |
| wrong_high_conf | $\text{wrong} \wedge (\text{completion\_bpt} \leq \tau_{\text{low\_wrong}})$ | Penalize confidently wrong. |
| ood | $(\text{prompt\_bpt} \geq \tau_{\text{prompt}})$ | Out-of-distribution prompt. |
| ind | $(\text{prompt\_bpt} < \tau_{\text{prompt}})$ | In-distribution prompt. |
| economical | $(\text{token\_ratio} \leq \text{len\_p33})$ | Short completions. |
| moderate | $(\text{len\_p33} < \text{token\_ratio} \leq \text{len\_p66})$ | Medium-length completions. |
| verbose | $(\text{token\_ratio} > \text{len\_p66})$ | Long completions. |
| boxed | $(\text{has\_box} \geq 0.5)$ | Answer is boxed. |
| unboxed | $(\text{has\_box} < 0.5)$ | Answer is not boxed. |
| A_high | $(compass\_context\_A \geq 0.6)$ | Verifier confidence high. |
| $\tau_{\text{high}}$ | 40th percentile of completion_bpt | |
| $\tau_{\text{low\_wrong}}$ | 60th percentile of completion_bpt among wrong items | |
| $\tau_{\text{prompt}}$ | 90th percentile of prompt_bpt | |
| len_p33, len_p66 | 33rd and 66th percentiles of token_ratio | |
| corr_p33, corr_p66 | 33rd and 66th percentiles of completion_bpt correct items | |

- **Conf-Wrong Penalty**:

$$\text{cat} = \begin{cases} 0 & \text{invalid} \\ 1 & \text{wrong}_{\text{high\_conf}} \\ 2 & \text{wrong} \wedge \text{low\_conf} \\ 3 & \text{correct} \end{cases}$$

Rubric weights $\mathbf{w}$ are chosen to reflect evaluation preferences. For example, *Format Aware* might use $[0, 0, 1, 2, 3]$ to mildly reward formatting when correct and slightly penalize confidently wrong (via schema choice); *Efficiency-Adjusted* can downweight verbose outputs among both correct and wrong categories.

- **Exact Match**  Correctness only; ignores formatting, confidence, and length.
- **Format Aware**  Rewards boxed, well-formatted answers; distinguishes boxed/unboxed even when wrong.
- **Conf-Calibrated**  Penalizes *confidently wrong*; grades correct answers by confidence (low/mid/high).
- **OOD Robustness**  Separates in-distribution vs. OOD prompts; checks correctness under both.
- **Strict Compliance**  Requires boxed final answers; unboxed-correct is treated as non-compliant.
- **Conf-Wrong Penalty**  Heavier penalty for wrong answers at high confidence; lighter when uncertain.
- **Verifier-Only**  Uses verifier signals alone to rank; model-agnostic prob of the verifier.
- **Format+Confidence**  Balanced composite over (boxed/unboxed) $\times$ (low/high confidence) for both wrong and correct; emphasizes boxed, high-confidence correctness and penalizes confidently wrong.

Table 5: Definitions of the twelve categorical evaluation schemata used in our Dirichlet–multinomial framework. Each schema specifies decision rules over correctness, formatting (boxed/unboxed), confidence (via completion_bpt), prompt distribution (in-distribution vs. OOD), output economy (via token_ratio), and verifier signals $(A, B, C)$. These rules map every attempt into $C+1$ discrete categories, enabling posterior means and credible intervals for any chosen weight vector $\mathbf{w}$.

| Categorical Schema | Rubric |
|---|---|
| Exact Match | 0 invalid; 1 wrong; 2 correct |
| Format Aware | 0 invalid; 1 wrong $\wedge$ unboxed; 2 wrong $\wedge$ boxed; 3 correct $\wedge$ unboxed; 4 correct $\wedge$ boxed |
| Conf-Calibrated | 0 invalid; 1 wrong $\wedge$ low_conf; 2 wrong_high_conf; 3 correct $\wedge$ low_conf; 4 correct $\wedge$ mid; 5 correct $\wedge$ high_conf |
| OOD Robustness | 0 invalid; 1 ood $\wedge$ wrong; 2 ind $\wedge$ wrong; 3 ood $\wedge$ correct; 4 ind $\wedge$ correct |
| Strict Compliance | 0 invalid; 1 wrong $\vee$ (correct $\wedge$ unboxed); 2 correct $\wedge$ boxed |
| Conf-Wrong Penalty | 0 invalid; 1 wrong_high_conf; 2 wrong $\wedge$ low_conf; 3 correct |
| Verifier-Only | 0 invalid; 1 high C; 2 high B; 3 A_high |
| Format+Confidence | 0 invalid; 1 wrong $\wedge$ unboxed; 2 wrong $\wedge$ boxed $\wedge$ low_conf; 3 wrong $\wedge$ boxed $\wedge$ high_conf; 4 correct $\wedge$ unboxed $\wedge$ low_conf; 5 correct $\wedge$ unboxed $\wedge$ high_conf; 6 correct $\wedge$ boxed $\wedge$ low_conf; 7 correct $\wedge$ boxed $\wedge$ high_conf |
| Length-Robust | 0 invalid; 1 wrong; 2 correct |
| Verifier Prob | 0 invalid; 1 wrong $\wedge$ A_high; 2 wrong $\wedge$ $\neg$ A_high; 3 correct $\wedge$ $\neg$ A_high; 4 correct $\wedge$ A_high |
| Efficiency-Adjusted | 0 invalid; 1 wrong $\wedge$ economical; 2 wrong $\wedge$ moderate; 3 wrong $\wedge$ verbose; 4 correct $\wedge$ economical; 5 correct $\wedge$ moderate; 6 correct $\wedge$ verbose |
| Concision-High-Conf | 0 invalid; 1 wrong; 2 correct $\wedge$ verbose; 3 correct $\wedge$ moderate; 4 correct $\wedge$ economical; 5 correct $\wedge$ economical $\wedge$ high_conf |

- **Length-Robust**   Isolates correctness irrespective of verbosity; does not penalize length.
- **Verifier Prob**   Probes agreement with the verifier: flags wrong with high verifier $A$ as inconsistent and distinguishes under/over-confidence on correct.
- **Efficiency-Adjusted**   Rewards short, correct completions; penalizes verbose outputs (especially when wrong).
- **Concision-High-Conf**   Prefers concise, high-confidence correct answers; downweights verbose correctness.

Figure 7 summarizes aggregated results across tasks. The leader 🔷 Qwen3-30B-A3B-Thinking ranks first under all selected schemata, but the margin to rank 2 depends on the rubric: it is largest under *Conf-Wrong Penalty* and smallest under *Verifier-Only*. Mid-pack reorderings are rubric-sensitive: under *Verifier Probe*, 📚 *OpenThinker2-32B* edges 📚 *gpt-oss-20b_medium*; under calibration-heavy schemata (e.g., *Conf-Calibrated*, *Format+Confidence*), 📚 *gpt-oss-20b_high* overtakes 📚 *OpenThinker2-32B*; *OOD Robustness* narrows the gap between ranks 2 and 3. Several categories (*Format Aware*, *Length-Robust*, *Strict Compliance*) agree closely, indicating that once correctness is accounted for, formatting and length rarely flip top ranks. In contrast, calibration-focused categories emphasize and penalize confidently wrong behavior, and efficiency-oriented categories favor concision. The lower tier is stable across categories (🪁 *EXAONE-4.0-1.2B*, 📚 *OpenThinker3-1.5B*, 📚 *OpenReasoning-Nemotron-1.5B*, 📘 *Sky-T1-32B-Flash*, 🐋 *DeepSeek-R1-Distill-Qwen-1.5B*), suggesting rubric choice primarily reshuffles the middle while preserving extremes. Overall, the categorical schemata surface complementary facets—format compliance, calibration, efficiency, OOD robustness, and verifier alignment—making rubric-dependent differences explicit and enabling compute-efficient, uncertainty-aware comparisons aligned with evaluation goals.

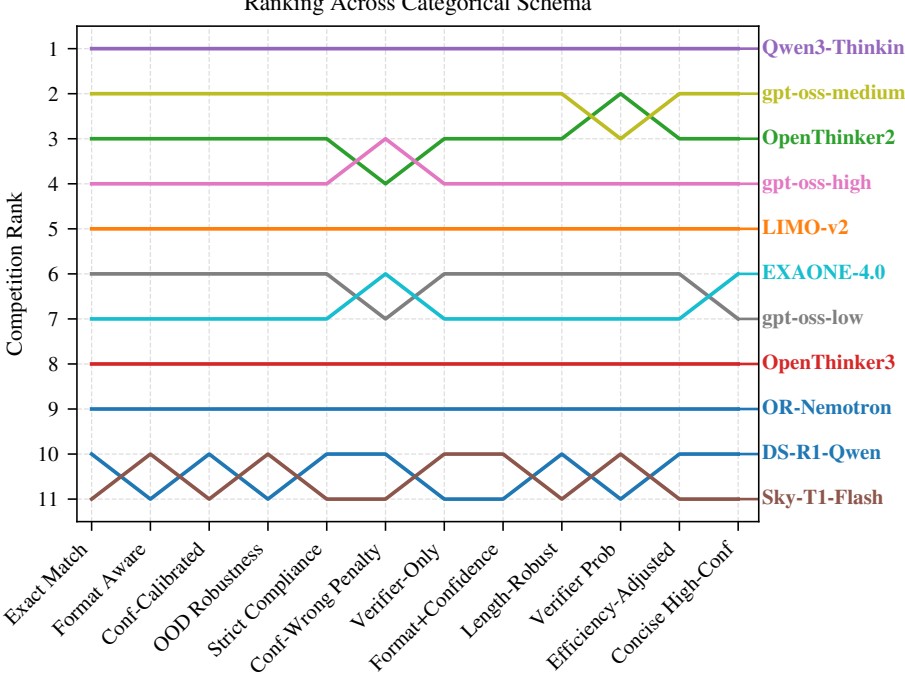

Figure 7: Competition ranks by model across selected categorical schema. Each column is a combination of base signals; lines indicate how a model's relative position shifts when the rubric changes.

## F.2 DOMAIN-AGNOSTIC RUBRIC-AWARE BAYES@N

The Bayesian construction is intentionally domain-agnostic: it applies whenever model outputs can be mapped into a finite set of categories equipped with a rubric. The evaluator specifies

1. a mapping from raw outputs (and any side information) to categorical labels $R_{\alpha i} \in \{0, \ldots, C\}$, and
2. a weight vector $w$ that encodes how those categories are valued.

Given these choices, Bayes@N returns the posterior mean $\mu(R)$ as a rubric-aware point estimate, and $\sigma(R)$ as an uncertainty estimate, for *any* such categorical evaluation.

This viewpoint naturally covers subjective tasks. For instance:

- In summarization, each response could be rated {bad, okay, good, excellent} or by multi-criteria scores such as faithfulness, coverage, style, and harmful content. Each discrete level becomes a category index $k$, and $w_k$ reflects the importance of that level or criterion.
- In dialogue safety, categories might distinguish {unsafe, borderline, safe}, or finer-grained notions such as policy violations vs. merely over-cautious refusals.

Once the labels are available (from humans or an LLM-as-a-judge), Bayes@N provides Bayesian estimates and credible intervals for any chosen rubric-based score, reusing the same closed-form posterior as in the binary case.

Two aspects are particularly promising for future work in such subjective domains:

1. **Preference-based evaluation with rubrics.** When model comparisons are driven by preferences (either from human experts or LLM judges), each comparison can be converted into categorical labels over rubric dimensions (e.g., faithfulness, verbosity, harmfulness). A downstream weight vector $w$ can then fold these dimensions into a single scalar score that reflects application-specific trade-offs.

2. **Transferring prior evidence across related tasks.** The optional prior matrix $R_0$ in Algorithm 1 lets us encode earlier outcome frequencies as a Dirichlet prior. For example, if a summarization system has been evaluated on a news dataset, the empirical category counts on that dataset can serve as prior counts when evaluating a closely related dataset. This allows stable rubric distributions to be reused across adjacent tasks or benchmark revisions, while still updating with new data.

An important limitation in subjective settings is that Bayes@N does *not* resolve disagreement or bias in the rubric or labeling process itself. The framework assumes a labeling scheme (from humans or an LLM-based judge) and a weight vector $w$ are given; it then provides a statistically principled way to aggregate those labels and quantify uncertainty. Designing good rubrics and calibrating judges remain separate modeling decisions.

## G  SCORIO

Alongside this paper, we release `Scorio`, an open-source Python package that implements the evaluation framework presented in this work. `Scorio` provides a simple, unified API for computing Bayes@$N$, avg@$N$, Pass@$k$, and their credible intervals, enabling researchers to adopt principled Bayesian evaluation with minimal effort. The package is available on PyPI and its documentation is hosted at `https://scorio.readthedocs.io`.

**Installation.**  `Scorio` can be installed via:

```
pip install scorio
```

**Basic usage.**  All evaluation functions operate on a results matrix $R \in \{0, \dots, C\}^{M \times N}$, where $M$ is the number of problems, $N$ is the number of trials per problem, and $C + 1$ is the number of outcome categories. An optional weight vector $w$ of length $C + 1$ maps each category to a score. Listing 1 shows binary evaluation using both the Bayesian estimator and Pass@$k$.

Listing 1: Binary evaluation with `Scorio`.

```python
import numpy as np
from scorio import eval

# Binary outcomes: M=2 problems, N=5 trials each
R = np.array([[0, 1, 1, 0, 1],
              [1, 1, 0, 1, 1]])

# Bayesian evaluation (binary: w defaults to (0, 1))
mu, sigma = eval.bayes(R)
print(f"Bayes@5: mu={mu:.4f}, sigma={sigma:.4f}")

# Average accuracy
a, sigma_a = eval.avg(R)
print(f"avg@5:   mu={a:.4f}, sigma={sigma_a:.4f}")

# Pass@k
print(f"Pass@1 = {eval.pass_at_k(R, k=1):.4f}")
print(f"Pass@2 = {eval.pass_at_k(R, k=2):.4f}")
```

**Credible intervals.**  Each estimator has a companion `_ci` function that returns the posterior mean, standard deviation, and a credible interval, as shown in Listing 2.

Listing 2: Computing credible intervals.

```python
# 95% credible interval for Bayes@N
mu, sigma, lo, hi = eval.bayes_ci(R, confidence=0.95)
print(f"Bayes@5: {mu:.4f} [{lo:.4f}, {hi:.4f}]")

```

```
5 # 95% credible interval for Pass@k
6 mu, sigma, lo, hi = eval.pass_at_k_ci(R, k=1)
7 print(f"Pass@1: {mu:.4f} [{lo:.4f}, {hi:.4f}]")
```

**Categorical (rubric-based) evaluation.** For graded outcomes with $C > 1$ categories, a weight vector $w$ specifies the score associated with each category. Listing 3 illustrates evaluation under a three-level rubric ($C = 2$) with partial credit.

Listing 3: Categorical evaluation with a weighted rubric.

```
1 # Graded outcomes: 0=incorrect, 1=partial, 2=correct
2 R = np.array([[0, 2, 1, 0, 2],
3               [2, 1, 1, 2, 1]])
4
5 # Weight vector: incorrect=0, partial=0.5, correct=1
6 w = np.array([0.0, 0.5, 1.0])
7
8 mu, sigma = eval.bayes(R, w)
9 print(f"Bayes@5 (graded): mu={mu:.4f}, sigma={sigma:.4f}")
10
11 mu, sigma, lo, hi = eval.bayes_ci(R, w, confidence=0.95)
12 print(f"95% CrI: [{lo:.4f}, {hi:.4f}]")
```

**Incorporating prior evidence.** When prior evaluation data are available (e.g., from a previous benchmark or a pilot study), they can be passed as a prior matrix $R^0$ to inform the posterior, as described in Section C. Listing 4 shows a minimal example in which a short pilot run is reused as prior evidence for a new evaluation.

Listing 4: Using prior evidence.

```
1 # Prior outcomes from a pilot study (M=2, D=3 trials)
2 R0 = np.array([[1, 0, 1],
3                [0, 1, 0]])
4
5 mu, sigma = eval.bayes(R, w=None, R0=R0)
6 print(f"Bayes@5 (with prior): mu={mu:.4f}, sigma={sigma:.4f}")
```

Table 6 summarizes the main `Scorio` API.

Table 6: Summary of the `Scorio` evaluation API. All functions accept a results matrix $R \in \{0, \ldots, C\}^{M \times N}$. Functions with the `_ci` suffix additionally return a credible interval.

| Function | Returns | Description |
|---|---|---|
| `bayes(R, w, R0)` | $(\mu, \sigma)$ | Bayesian posterior mean and uncertainty |
| `bayes_ci(R, w, R0)` | $(\mu, \sigma, \text{lo}, \text{hi})$ | + credible interval |
| `avg(R, w)` | $(a, \sigma_a)$ | Weighted average and uncertainty |
| `avg_ci(R, w)` | $(a, \sigma_a, \text{lo}, \text{hi})$ | + credible interval |
| `pass_at_k(R, k)` | $p$ | Pass@$k$ estimate |
| `pass_at_k_ci(R, k)` | $(\mu, \sigma, \text{lo}, \text{hi})$ | + credible interval |
| `pass_hat_k(R, k)` | $p$ | Pass^$k$ |
| `g_pass_at_k_tau(R, k, tau)` | $p$ | G-Pass@$k_\tau$ |
| `mg_pass_at_k(R, k)` | $p$ | mG-Pass@$k$ |

## H EXTENDED RELATED WORK

The evaluation of LLMs in generative reasoning tasks, under test-time scaling (e.g., via repeated sampling(53)), has evolved to address the stochastic nature of inference and the need for robust measures of functional correctness. Early approaches relied on syntactic similarity metrics like

BLEU (54) and CodeBLEU (55), which compare generated answers against reference solutions. However, these metrics often fail to capture semantic correctness in reasoning tasks, motivating metrics based on execution-validation or test-based validation (56; 55). This limitation has shifted focus toward functional evaluation, where the generated solution is assessed via a ground truth to verify correctness(56; 57). In this section, we review key functional metrics, focusing on those that leverage multiple samples to scale performance at inference time. These metrics form the basis to assess LLM capabilities but often overlook probabilistic uncertainty or consistency across samples, motivating our novel Bayesian framework.

**The Pass@$k$ metric** was originally introduced by (56; 24) for evaluating LLMs trained on code. It measures the probability that at least one of $k$ independently generated samples for a given problem passes all associated unit tests (i.e., by matching ground-truth answers or satisfying logical constraints), offering a practical estimate of a model's potential performance in solving a variety of complex tasks and problems. The unbiased estimator of Pass@$k$ is computed as:

$$\text{Pass@}k = \mathbb{E}_{\text{problems}} \left[ 1 - \frac{\binom{n-c}{k}}{\binom{n}{k}} \right], \tag{21}$$

where $n$ is the total number of generated samples and $c$ is the total number of correct solutions within the $n$ trials. This estimator has smaller uncertainty in the limit of $n \gg k$, ensuring reliable approximations. However, due to computational costs, $k$ is often comparable to $n$ in practice, which can increase variance and weaken evaluation stability. The Pass@$k$ metric has been adapted beyond code to evaluate LLMs in various tasks requiring verifiable correctness, such as math, logic, and general reasoning (57; 58; 59; 60).

Although Pass@$k$ was initially introduced in the context of coding, it later became the de facto choice to evaluate LLMs not only on math reasoning tasks (45; 46; 47; 61; 62; 63; 64; 65; 27; 66) but also on safety evaluations spanning agent red-teaming, jailbreaks, and backdoor analyses (67; 68; 69; 70; 71; 72). This broad adoption makes the metric's instability consequential: when rankings are based on limited trials or small benchmarks, variance in Pass@$k$-style estimates can affect conclusions about model progress and deployment readiness.

**Pass^$k$**, introduced in (48), extends the Pass@$k$ metric to capture both the potential performance and the consistency of LLMs in reasoning tasks, where evaluating the reliability and stability of generated solutions is crucial. Pass^$k$ is defined as the probability that all $k$ trials are correct:

$$\text{Pass^}k = \mathbb{E}_{\text{problems}} \left[ \frac{\binom{c}{k}}{\binom{n}{k}} \right], \tag{22}$$

where $c$ and $n$ retain the same meanings as in Pass@$k$. This metric assumes that all the trials are independent and uniformly distributed, approximating the binomial distribution with a hypergeometric distribution to account for sampling without replacement. By requiring all $k$ samples to be correct, Pass^$k$ provides a stringent measure of model consistency and stability.

To introduce flexibility, Liu et al. (27) proposed **G-Pass@$k_{\tilde{\tau}}$**, which incorporates a tolerance threshold $\tilde{\tau} \in (0.0, 1.0]$:

$$\text{G-Pass@}k_{\tilde{\tau}} = \mathbb{E}_{\text{problems}} \left[ \sum_{j=\lceil \tau \cdot k \rceil}^{c} \frac{\binom{c}{j} \cdot \binom{n-c}{k-j}}{\binom{n}{k}} \right], \tag{23}$$

where $\lceil \tau \cdot k \rceil$ is the smallest integer greater than or equal to $\tau \cdot k$. This formulation allows up to $k - \lceil \tau \cdot k \rceil$ incorrect solutions, balancing the assessment of potential with consistency. As a special case, Pass@$k$ corresponds to G-Pass@$k_{\tilde{\tau}}$ in the limit $\tau \to 0$.

Furthermore, Liu et al. (27) introduced **mG-Pass@$k$**, an interpolated metric that integrates G-Pass@$k_{\tilde{\tau}}$ over $\tau \in [0.5, 1.0]$:

$$\text{mG-Pass@}k = 2 \int_{0.5}^{1.0} \text{G-Pass@}k_{\tau} d\tau \approx \frac{2}{k} \sum_{i=\lceil 0.5 \cdot k \rceil + 1}^{k} \text{G-Pass@}k_{i/k}, \tag{24}$$

providing a more comprehensive measure that jointly reflects performance potential and reasoning stability.

These extended metrics have been applied to mathematical reasoning benchmarks such as Live-MathBench, MATH, and AIME, where they reveal substantial performance degradation of LLMs under stricter stability requirements.

**Evaluation infrastructure and reporting practice.** Efforts like HELM advance holistic, transparent evaluation across scenarios and metrics (5), while practice guidelines distill reproducibility pitfalls and prescribe multi-run, uncertainty-aware reporting with fixed prompts, decoding, and dataset/version control (49). The LM Evaluation Harness offers standardized, reproducible frameworks to implement these recommendations (49). It supports uncertainty reporting through binomial-style uncertainty estimates for binary mean metrics and bootstrap estimates for others.

**Uncertainty-aware LLM evaluation.** Recent work increasingly emphasizes interval-aware, small-sample-valid reporting rather than CLT/Wald error bars. Bowyer et al. show that CLT-based intervals *miscalibrate* on small benchmarks and advocate small-$n$-appropriate frequentist or Bayesian intervals for reliable comparisons (42). A Bayesian alternative models capability as a latent success probability and reports posterior uncertainty that remains informative with limited trials, yielding more stable rankings (34). In judge-based settings, *Judging LLMs on a Simplex* places model and judge behavior on the probability simplex, enabling uncertainty-aware comparisons and highlighting how distributional structure matters for evaluation (50). Beyond bespoke LLM metrics, prediction-powered inference supplies general procedures for valid confidence intervals that leverage model predictions to reduce labeled-sample requirements (73). Finally, in adjacent retrieval evaluation with LLM-generated assessments, Oosterhuis et al. construct reliable confidence intervals and demonstrate that calibrated uncertainty, rather than point estimates, should guide decisions, reinforcing this shift for LLM evaluation more broadly (51).

# I EXPERIMENT SETUP AND REPRODUCIBILITY

## I.1 METRICS

**Kendall's Tau:** Kendall's tau ($\tau$) (74) is a nonparametric rank correlation coefficient that quantifies the ordinal relationship between two ranked sets by evaluating the consistency in their orderings. For two rankings of $n$ items, it examines all unique pairs $(i, j)$ where $i < j$:

- A pair is *concordant* if the relative ordering of items $i$ and $j$ is the same in both rankings (both place $i$ before $j$ or vice versa).
- A pair is *discordant* if the relative ordering is different.
- Pairs with ties in either ranking are neither concordant nor discordant.

Define $n_c$ as the number of concordant pairs, $n_d$ as the number of discordant pairs, and $n_0 = n(n-1)/2$ as the total number of unique pairs. Let $n_1$ represent the number of tied pairs in the first ranking, and $n_2$ similarly for the second ranking. The two common variants are the following:

$$\text{Tau-a:} \quad \tau_a = \frac{n_c - n_d}{n_0} \qquad \text{(no adjustment for ties)}, \tag{25}$$

$$\text{Tau-b:} \quad \tau_b = \frac{n_c - n_d}{\sqrt{(n_0 - n_1)(n_0 - n_2)}} \qquad \text{(adjusts for ties in both rankings)}. \tag{26}$$

Tau-a assumes no ties and may underestimate correlation when ties occur. Tau-b, which corrects for ties, is better suited for datasets with equivalent rankings.

In our implementation, we use `scipy.stats.kendalltau` with its default variant='b', which computes $\tau_b$ efficiently and handles ties appropriately. The coefficient ranges from $-1$ (perfect disagreement) to $+1$ (perfect agreement), with $0$ indicating no association. This metric provides a robust, distribution-free measure for comparing model performance rankings, particularly when ties reflect meaningful equivalences.

**Convergence@$n$.** For a given bootstrap replicate, we measure convergence in terms of an *exact ranking match*. At each step $s \in \{1, \ldots, N_{\max}\}$, we compute the ranking induced by the first $s$

trials and compare it to a gold-standard ranking (obtained from all $N_{\max}$ trials). We then define

$$s^{\star} \;=\; \min\Big\{ s \le N_{\max}-1 \;\Big|\; \begin{array}{l} \text{the ranking after } s \text{ trials matches the gold-standard ranking,} \\ \text{and remains unchanged after every subsequent trial} \end{array} \Big\},$$

and refer to $s^{\star}$ as the convergence@$n$ value for that replicate. If no such $s^{\star} \le N_{\max}$ exists, we declare that replicate to exhibit *no convergence*.

## I.2 MODELS AND DATASETS

**Datasets.** We evaluate on four mathematical reasoning test sets: AIME'24 (25), AIME'25 (26), BrUMO'25 (37), and HMMT'25 (36). AIME is administered by the Mathematical Association of America and consists of two sets of 15 integer-answer problems; we use the 2024 and 2025 problem sets. For HMMT'25, we use the officially posted February 2025 contest set (algebra, geometry, number theory, and combinatorics). For BrUMO'25, we use the published 2025 problem sets from the tournament archive.

**Models.** Unless noted otherwise, we run each generator with the provider-recommended chat template (DeepSeek/Qwen style when unspecified) and identical decoding settings (below) to minimize template-induced variance. The base model cohort includes 11 models (8 distinct models + 3 modes (low, medium, and high) of gpt-oss) as follows: Sky-T1-32B-Flash (75) (reasoning-optimized "flash" variant tied to overthinking-reduction work), Qwen3-30B-A3B-Thinking-2507 (76) (Qwen3 series, reasoning variant), DeepSeek-R1-Distill-Qwen-1.5B (45) (distilled reasoning model), gpt-oss-20b (77) (OpenAI open-weight reasoning model; we use the default quantization, MXFP4, and, for prompting, rely on OpenAI Harmony, which defines three levels of reasoning effort), LIMO-v2 (78) (data-efficient reasoning fine-tuned on curated traces), EXAONE-4.0-1.2B (79) (hybrid non-reasoning/reasoning modes), OpenReasoning-Nemotron-1.5B (80; 81; 82; 83) (open-weight small reasoning model), OpenThinker2-32B (84) and OpenThinker3-1.5B (84) (trained on OpenThoughts2/3 data recipes).

To investigate the effect of the number of models required to reach a stable ranking with and without credible intervals, in addition to the 11 above-mentioned models, we extend the evaluation to 20 models in total (17 + 3): Phi-4-reasoning and Phi-4-reasoning-plus (85) (14B small language models with supervised "teachable" reasoning traces and an RL-enhanced variant), OpenR1-Distill-7B (86) (an open 7B distillation of DeepSeek-R1 using fully public data), FuseO1-DeepSeekR1-QwQ-SkyT1-Flash-32B-Preview (87) (System-II "long-short" reasoning fusion of DeepSeek-R1, QwQ, and Sky-T1-32B-Flash), Light-R1-14B-DS (88) (a Qwen2.5-based long-chain-of-thought model further improved with GRPO-style reinforcement learning), AceReason-Nemotron-1.1-7B (89) (7B NVIDIA Nemotron math/code model trained on Open-MathReasoning/OpenCodeReasoning data), NVIDIA-Nemotron-Nano-9B-v2 (90) (a hybrid Mamba-Transformer "Nano 2" model with controllable reasoning mode), Qwen3-4B-Thinking-2507 (76) (4B "thinking" variant of Qwen3 with scaled reasoning depth), and Bespoke-Stratos-7B (91) (Qwen2.5-7B student obtained via DeepSeek-R1-based reasoning distillation on Bespoke-Stratos-17k).

For verification, we additionally use CompassVerifier-3B (92), a lightweight answer verifier suitable for outcome reward and equivalence checking.

**Prompting.** For most models, we follow the provider-recommended DeepSeek/Qwen-style prompt: *"Please reason step by step, and put your final answer within \boxed{}."* For gpt-oss-20b, we instead use the OpenAI Harmony prompt template, which provides three levels of reasoning effort. For OpenReasoning-Nemotron-1.5B, we adopt the task-specific prompt: *"Solve the following math problem. Make sure to put the answer (and only the answer) inside \boxed{}."*

## I.3 REPRODUCIBILITY

**Sampling setup.** All trials use top-$p$ sampling with temperature 0.6, $p = 0.95$, batch size 1, and seeds 1234–1313. We perform $N = 80$ trials per dataset $\times$ model.

| ID | Model | Short name |
|----|-------|-----------|
| 1 | DeepSeek-R1-Distill-Qwen-1.5B | DS-R1-Qwen |
| 2 | LIMO-v2 | LIMO-v2 |
| 3 | OpenThinker2-32B | OpenThinker2 |
| 4 | OpenThinker3-1.5B | OpenThinker3 |
| 5 | Qwen3-30B-A3B-Thinking-2507 | Qwen3-Thinking |
| 6 | Sky-T1-32B-Flash | Sky-T1-Flash |
| 7 | gpt-oss-20b_high | gpt-oss-high |
| 8 | gpt-oss-20b_low | gpt-oss-low |
| 9 | gpt-oss-20b_medium | gpt-oss-medium |
| 10 | EXAONE-4.0-1.2B | EXAONE-4.0 |
| 11 | OpenReasoning-Nemotron-1.5B | OR-Nemotron |
| 12 | Phi-4-reasoning | Phi-4 |
| 13 | Phi-4-reasoning-plus | Phi-4-plus |
| 14 | OpenR1-Distill-7B | OR1-Distill |
| 15 | FuseO1-DeepSeekR1-QwQ-SkyT1-Flash-32B-Preview | FuseO1-DS-QwQ-SkyT1 |
| 16 | Light-R1-14B-DS | Light-R1-DS |
| 17 | AceReason-Nemotron-1.1-7B | AR-Nemotron |
| 18 | NVIDIA-Nemotron-Nano-9B-v2 | NVIDIA-Nemotron |
| 19 | Qwen3-4B-Thinking-2507 | Qwen3-4B |
| 20 | Bespoke-Stratos-7B | Bespoke |

Table 7: Mapping between model IDs, full model names, and the shortened names used in figures and legends. Corresponding subsets are listed in Tables 9, 10, and 11.

**Verifier.** We use 🎓 `CompassVerifier-3B` as a reward model. During evaluation, we leverage the model's scores on prompts generated by other models to create categorical schemas. We rely on the `Transformers` (52) and `Accelerate` (93) libraries. To maximize throughput, we enable FlashAttention kernels (23) and adopt the `DFloat11` format (94).

**Serving stack.** Token generation is served with `vLLM` (PagedAttention) (16), and models are loaded in `bf16` unless the release requires MXFP4 (e.g., `gpt-oss`). We record log-probabilities for both the input prompt and generated tokens, and cap `max_tokens` at 32,768.

**Hardware.** All runs execute on clusters with $8\times$ NVIDIA H200 (141GB).

COMPUTATIONAL COST AND TOKEN STATISTICS

Across all tasks, we evaluated 20 models with 80 trials per model and 30 questions per benchmark, yielding a total of 192,000 independent inference runs. This required 7,445 GPU-hours ($\sim$310 GPU-days) and generated 2.96B tokens (2,963,318,176) in total (see Figure 8 for details).

| Task | Inference Time (hours) | Completion Tokens (M) |
|------|------------------------|-----------------------|
| AIME'24 | 1,699.4 | 680.0 |
| AIME'25 | 1,878.4 | 728.3 |
| HMMT'25 | 2,216.5 | 851.2 |
| BrUMO'25 | 1,650.9 | 666.9 |
| **TOTAL** | **7,445.2** | **2,926.4** |

Table 8: Task-level computational cost aggregated over 20 models, 80 trials, 4 tasks, and 30 questions per task. Token counts correspond to *completion* tokens only.

**Task-level computational cost.** HMMT'25 is the most expensive benchmark in terms of GPU time (2,217 GPU-hours), while BrUMO'25 is the least expensive (1,651 GPU-hours). Figure 8 provides a complementary visualization of these patterns, showing inference time and completion-token usage across models and tasks.

**Token breakdown.** Aggregating across all tasks and models, the total number of tokens (prompt + completion) is 2.96B. The breakdown is:

- **Prompt tokens:** 37M (1.2%)
- **Completion tokens:** 2.93B (98.8%)
- **Average per query:** 15,434 tokens

**GPU-hours by model efficiency.** The 20 model configurations varied substantially in computational efficiency:

- Most efficient: `gpt-oss-20b-low` (48.4 GPU-hours for 9,600 queries)
- Least efficient: `LIMO-v2` (894.3 GPU-hours for 9,600 queries)
- Average per query over all models: 139.6 seconds ($\sim$2.3 minutes)

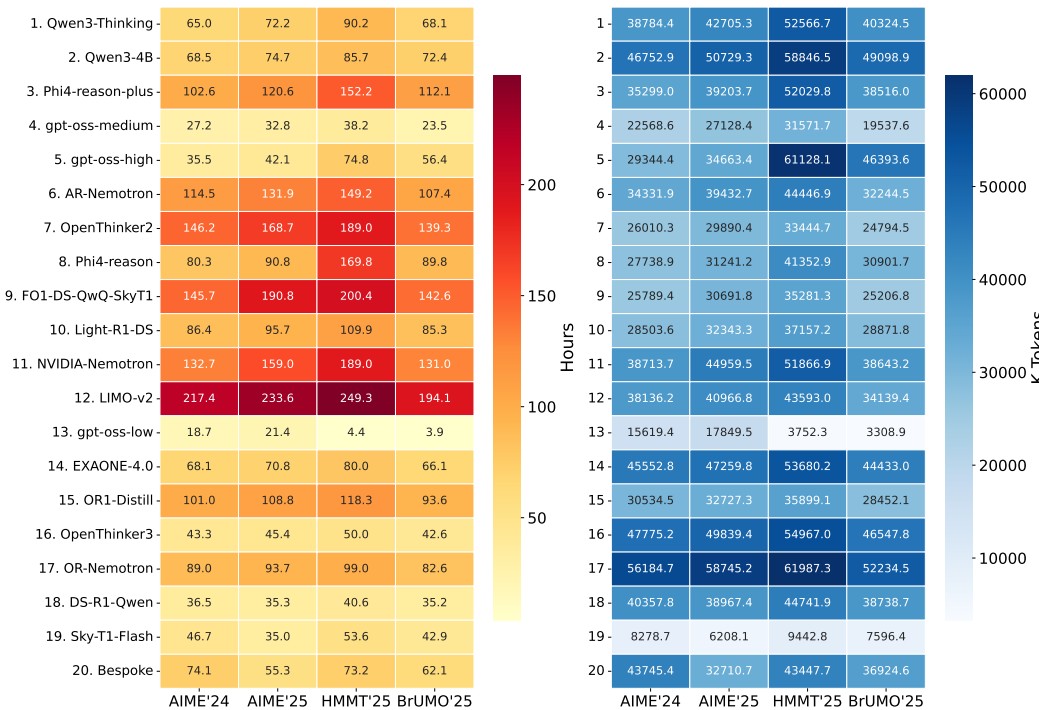

Figure 8: **Computational cost analysis.** (**Left**) Total inference time in hours aggregated over 80 trials and 30 questions per benchmark (2,400 inference runs per cell). (**Right**) Total number of completion tokens (in thousands) generated across the same runs. Models are ordered by overall performance (best to worst, top to bottom).

## J CONVERGENCE

While Figure 1 shows the PMF of convergence@$n$, Figure 9 shows the corresponding cumulative distribution functions (CDFs). For Pass@4 and Pass@8, there is no convergence, as the figure shows no CDFs associated with them. The CDFs are computed using the same bootstrap replicates as in Figure 1. The distribution of convergence@$n$ is computed using the result matrices $R$ from the first 11 models (Table 7). Among the $10^5$ replications, Figure 10 shows the worst-case scenarios in which convergence@$n$ attains its maximum value. As discussed in Section 3.3, convergence@$n$ depends on the number of models $L$: as $L$ increases, convergence@$n$ grows. When we extend the pool of LLMs from 11 to 20 models, convergence@$n$ reaches *no convergence* for all datasets (see Figure 11).

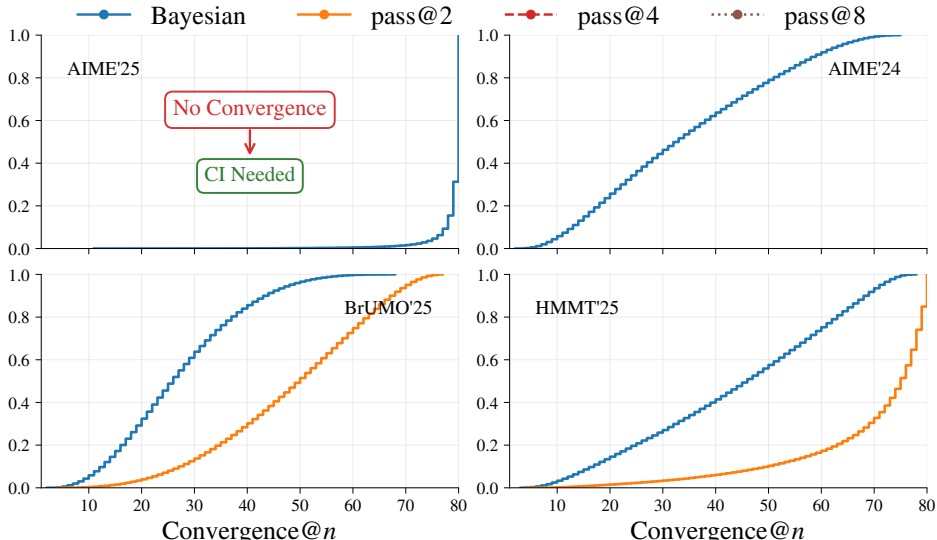

Figure 9: **CDF of convergence@$n$.** Complementing the PMFs in Figure 1, these CDFs plot $P(k \le n)$ for the convergence threshold $k$ across AIME'24, AIME'25, HMMT'25, and BrUMO'25. Steeper and earlier rises indicate faster convergence. Bayes@$N$ accumulates mass with fewer trials than Pass@2/4/8, and on AIME'24/'25 the Pass curves do not reach 1 by $N_{\max} = 80$. Greater convergence suggests that credible intervals should be reported for the evaluation tasks.

To complement the worst-case trajectories discussed in Section 3.3 and shown in Figures 10 and 11, we provide additional details on the construction of the model subsets and the resulting convergence behavior. Table 7 lists the pool of 20 LLMs used in this analysis, together with the shortened identifiers that appear throughout the figures and tables. From this pool we construct 50 subsets of 5 models, 20 subsets of 10 models, and 20 subsets of 15 models, as summarized in Tables 9 to 11. Each row in these tables corresponds to one subset, indicating which models are included and reporting, under each task, the convergence@$n$ metric computed *without* a credible interval; each entry is the mean over $10^5$ bootstrap replicates. Thus, the tables make explicit how convergence@$n$ depends not only on the task but also on the particular mixture of models being compared. Aggregating across all subsets and replicates, Figure 4 then visualizes the distribution of convergence@$n$ as a function of the number of models $L$, confirming the trend anticipated in the main text: as $L$ grows from 5 to 15 and ultimately to the full set of 20 LLMs, the required number of trials increases and non-convergence becomes common, indicating that rank-based evaluation methods such as avg@$N$ and the Pass@$k$ family become increasingly unreliable without an accompanying Bayesian uncertainty quantification such as Bayes@$N$.

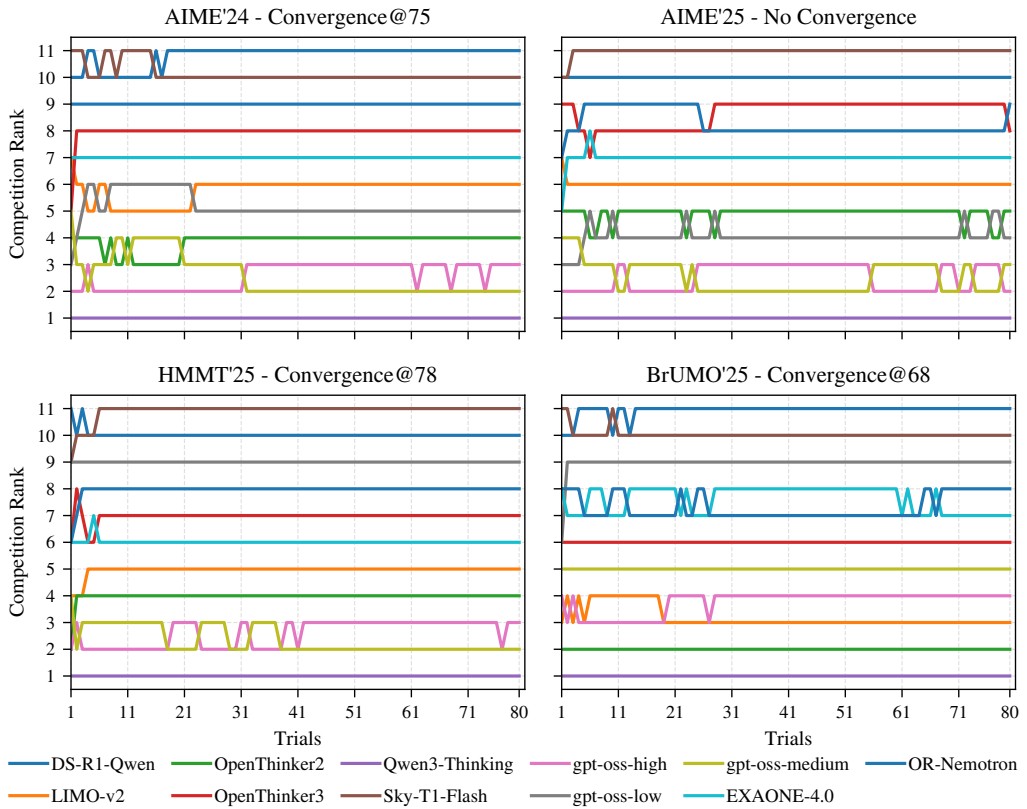

Figure 10: **Worst-case bootstrap rank trajectories**. Each line shows the ranking of a model as trials are added (11 models in total). Convergence is defined as the minimal $N$ after which the ranking remains unchanged. (a) AIME'24: converges at $N = 75$. (b) AIME'25: no convergence observed within 80 trials. (c) HMMT'25: converges at $N = 78$. (d) BrUMO'25: converges at $N = 68$.

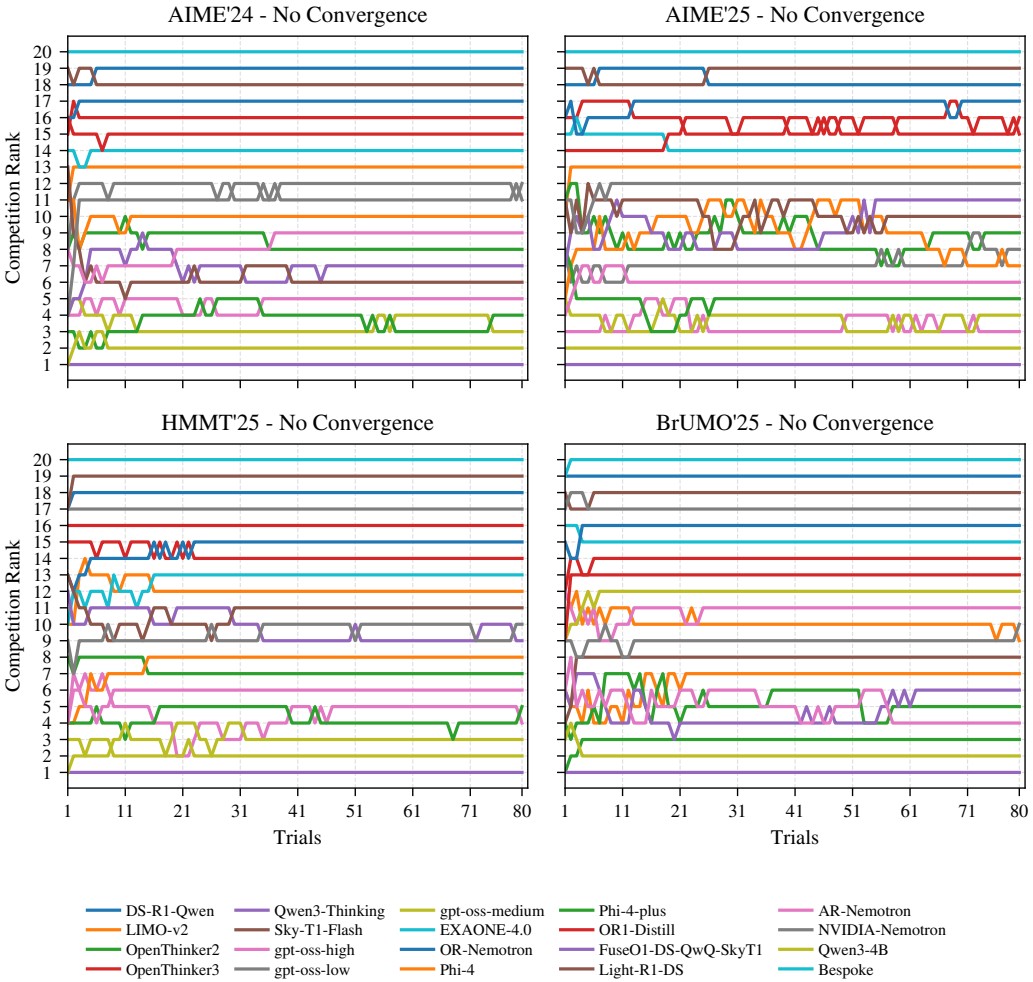

Figure 11: **Worst-case bootstrap rank trajectories**. Each line shows the ranking of a model as trials are added (20 models in total). Convergence is defined as the minimal $N$ after which the ranking remains unchanged. There is at least one *no convergence* replicate among the $10^5$ bootstrapped replications.

Table 9: **5-model combinations**. Matrix showing model presence across the 50 evaluated combinations. Values under each task report the convergence@$n$ metric computed without a credible interval; each value is the mean of $10^5$ bootstrapped samples. Model identifiers are listed in Table 7.

| Comb. | 1 | 2 | 3 | 4 | 5 | 6 | 7 | 8 | 9 | 10 | 11 | 12 | 13 | 14 | 15 | 16 | 17 | 18 | 19 | 20 | AIME'24 | AIME'25 | HMMT'25 | BrUMO'25 |
|---|---|---|---|---|---|---|---|---|---|---|---|---|---|---|---|---|---|---|---|---|---|---|---|---|
| 1 | | | | 4 | | | | | | | | | 13 | 14 | | | | 18 | 19 | | 13.5 | 69.0 | 11.9 | 8.8 |
| 2 | 1 | 2 | | | | | | | | | | | | 14 | | | | 18 | | 20 | 3.0 | 38.4 | 13.3 | 59.3 |
| 3 | 1 | 2 | | 4 | | | | | | | | | 13 | | | | | | | 20 | 1.5 | 3.1 | 2.7 | 2.7 |
| 4 | | | | | | | | | | | 11 | | 13 | 14 | | | | 18 | 19 | | 10.9 | 65.0 | 12.6 | 5.0 |
| 5 | 1 | | | | | | | | | | 11 | | 13 | | | | 17 | | | 20 | 2.5 | 5.5 | 3.7 | 50.2 |
| 6 | 1 | | | | | | | | | 10 | | | | 14 | | | 17 | 18 | | | 11.4 | 15.3 | 7.9 | 3.1 |
| 7 | 1 | | | | | | | | | 10 | | 12 | 13 | | | | | | 19 | | 11.0 | 3.3 | 12.5 | 11.1 |
| 8 | 1 | | | 4 | | | | | | 10 | | | | | | | 17 | 18 | | | 5.9 | 13.1 | 9.8 | 6.3 |
| 9 | | | | | | | | | | | 11 | 12 | | | | | 17 | | 19 | 20 | 44.3 | 2.8 | 5.1 | 9.4 |
| 10 | 1 | 2 | | | | | | | | | | | 13 | | | | 17 | 18 | | | 6.8 | 38.8 | 14.6 | 68.4 |
| 11 | | | | | | | | | | 10 | 11 | | | 14 | | | 17 | | 19 | | 10.4 | 66.9 | 5.3 | 16.6 |
| 12 | | | | | | | | | | | 11 | | 13 | 14 | | | 17 | | | 20 | 3.7 | 65.1 | 5.4 | 50.3 |
| 13 | | | | 4 | | | | | | 10 | 11 | | | | | | | 18 | 19 | | 9.2 | 72.0 | 17.2 | 17.7 |
| 14 | 1 | 2 | | | | | | | | | | 12 | 13 | | | | | | | 20 | 2.9 | 4.4 | 5.8 | 11.5 |
| 15 | | | | 4 | | | | | | | 11 | 12 | | 14 | | | | | | 20 | 12.9 | 79.0 | 15.0 | 8.4 |
| 16 | | 2 | | | | | | | | 10 | 11 | | | | | | | 18 | 19 | | 4.7 | 41.1 | 27.4 | 60.3 |
| 17 | 1 | 2 | | 4 | | | | | | | | | | | | | 17 | | | 20 | 1.8 | 3.2 | 3.1 | 2.3 |
| 18 | 1 | 2 | | 4 | | | | | | | | | | | | | 17 | 18 | | | 6.2 | 38.3 | 14.3 | 59.4 |
| 19 | 1 | 2 | | | | | | | | | | 12 | | | | | 17 | | 19 | | 44.3 | 4.2 | 8.1 | 11.1 |
| 20 | 1 | | | | | | | | | 10 | 11 | | | | | | | 18 | | 20 | 1.8 | 13.1 | 7.8 | 15.0 |
| 21 | 1 | | | | | | | | | 10 | | 12 | | | | | | 18 | | 20 | 5.4 | 4.1 | 16.4 | 3.1 |
| 22 | | 2 | | | | | | | | | | 12 | | 14 | | | 17 | | | 20 | 44.3 | 4.1 | 7.9 | 9.8 |
| 23 | | | | 4 | | | | | | | 11 | | 13 | | | | 17 | | 19 | | 14.7 | 71.2 | 19.7 | 50.9 |
| 24 | 1 | 2 | | 4 | | | | | | | | | | 14 | | | | 18 | | | 8.4 | 72.3 | 14.0 | 59.8 |
| 25 | | | | 4 | | | | | | 10 | | 12 | | | | | | 18 | | 20 | 6.3 | 13.6 | 17.1 | 7.0 |
| 26 | 1 | | | 4 | | | | | | | | | | | | | | 18 | 19 | 20 | 1.4 | 2.9 | 2.0 | 1.8 |
| 27 | | | | | | | | | | 10 | | | | 14 | | | 17 | | 19 | 20 | 9.5 | 15.3 | 2.5 | 5.2 |
| 28 | | | | 4 | | | | | | | | | | | | | 17 | 18 | 19 | 20 | 5.4 | 1.6 | 3.3 | 4.5 |
| 29 | | | | | | | | | | 10 | | 12 | | | | | 17 | 18 | | 20 | 45.1 | 4.1 | 17.5 | 8.4 |
| 30 | | 2 | | | | | | | | 10 | 11 | | | | | | 17 | 18 | | | 7.3 | 41.1 | 27.5 | 60.3 |
| 31 | | | 3 | | 5 | | 7 | | 9 | | | | | | | 16 | | | | | 39.6 | 73.6 | 38.1 | 13.1 |
| 32 | | | 3 | | | 6 | | 8 | | | | | | | 15 | 16 | | | | | 48.0 | 71.4 | 32.9 | 18.6 |
| 33 | | | | | 5 | 6 | 7 | 8 | | | | | | | | 16 | | | | | 19.5 | 39.5 | 1.6 | 10.4 |
| 34 | | | 3 | | 5 | | | 8 | 9 | | | | | | | 16 | | | | | 23.9 | 67.8 | 6.5 | 3.3 |
| 35 | | | | | | 6 | 7 | 8 | 9 | | | | | | | 16 | | | | | 35.7 | 73.1 | 36.9 | 17.1 |
| 36 | | | 3 | | 5 | 6 | 7 | 8 | | | | | | | | | | | | | 10.2 | 61.1 | 2.3 | 10.2 |
| 37 | | | | | 5 | | 7 | | 9 | | | | | | 15 | 16 | | | | | 47.3 | 74.7 | 47.4 | 16.6 |
| 38 | | | 3 | | 5 | | 7 | 8 | 9 | | | | | | | | | | | | 29.6 | 75.4 | 37.1 | 12.1 |
| 39 | | | | | 5 | 6 | | | 9 | | | | | | 15 | 16 | | | | | 35.6 | 47.0 | 28.6 | 10.1 |
| 40 | | | 3 | | | 6 | | 8 | 9 | | | | | | | 16 | | | | | 23.9 | 67.8 | 6.4 | 11.0 |
| 41 | | | 3 | | | 6 | 7 | 8 | | | | | | | 15 | | | | | | 36.0 | 64.3 | 10.9 | 13.9 |
| 42 | | | | | | 6 | 7 | 8 | | | | | | | 15 | 16 | | | | | 40.5 | 59.1 | 28.6 | 15.8 |
| 43 | | | 3 | | 5 | 6 | | | | | | | | | 15 | 16 | | | | | 47.8 | 60.1 | 32.9 | 14.7 |
| 44 | | | 3 | | 5 | | 7 | | 9 | | | | | | 15 | | | | | | 43.0 | 72.5 | 39.1 | 16.0 |
| 45 | | | | | 5 | | 7 | 8 | 9 | | | | | | 15 | | | | | | 31.3 | 72.6 | 36.9 | 12.1 |
| 46 | | | 3 | | 5 | 6 | | 8 | | | | | | | | 16 | | | | | 19.8 | 67.8 | 6.0 | 11.0 |
| 47 | | | 3 | | 5 | 6 | | 8 | | | | | | | 15 | | | | | | 33.0 | 64.3 | 10.3 | 13.9 |
| 48 | | | 3 | | | 6 | 7 | | 9 | | | | | | | 16 | | | | | 39.6 | 73.6 | 38.1 | 13.1 |
| 49 | | | | | 5 | 6 | 7 | | | | | | | | 15 | 16 | | | | | 39.9 | 47.0 | 28.6 | 10.5 |
| 50 | | | 3 | | | 6 | 7 | 8 | | | | | | | | 16 | | | | | 29.8 | 67.8 | 6.7 | 11.2 |

Table 10: **10-model combinations**. Matrix showing model presence across the 20 evaluated combinations. Values under each task report the convergence@$n$ metric computed without a credible interval; each value is the mean of $10^5$ bootstrapped samples. Model identifiers are listed in Table 7.

| Comb. | 1 | 2 | 3 | 4 | 5 | 6 | 7 | 8 | 9 | 10 | 11 | 12 | 13 | 14 | 15 | 16 | 17 | 18 | 19 | 20 | AIME'24 | AIME'25 | HMMT'25 | BrUMO'25 |
|---|---|---|---|---|---|---|---|---|---|---|---|---|---|---|---|---|---|---|---|---|---|---|---|---|
| 1 | 1 | 2 | 3 | 4 | 5 | 6 | | | | | | | | 14 | | 16 | | | 19 | 20 | 28.2 | 71.9 | 23.3 | 35.4 |
| 2 | 1 | 2 | | 4 | | | | | | | 11 | | | 14 | | 16 | 17 | 18 | 19 | 20 | 16.4 | 79.0 | 43.2 | 60.4 |
| 3 | | | 3 | | 5 | 6 | 7 | 8 | 9 | | | 12 | 13 | | 15 | 16 | | | | | 68.4 | 79.3 | 71.3 | 73.5 |
| 4 | | 2 | | | | | 7 | 8 | 9 | | 11 | 12 | 13 | | | 16 | 17 | 18 | | | 76.5 | 79.0 | 70.3 | 74.8 |
| 5 | 1 | 2 | | 4 | | | | | | 10 | 11 | | | 14 | | | 17 | 18 | 19 | 20 | 18.1 | 79.0 | 30.6 | 60.9 |
| 6 | 1 | 2 | | 4 | | | | | | | 11 | 12 | 13 | | | 16 | 17 | | 19 | 20 | 47.9 | 72.8 | 30.5 | 70.0 |
| 7 | | | 3 | | 5 | 6 | 7 | 8 | | | | 12 | 13 | 14 | 15 | 16 | | | | | 56.6 | 78.3 | 68.7 | 73.5 |
| 8 | 1 | | 3 | 4 | 5 | 6 | 7 | 8 | 9 | | | | | | 15 | | | | 19 | | 46.3 | 75.8 | 49.4 | 38.6 |
| 9 | 1 | 2 | | 4 | | | | | | 10 | 11 | 12 | 13 | | | 16 | | 18 | 19 | | 20.8 | 74.4 | 48.1 | 73.0 |
| 10 | | | | | 5 | | 7 | 8 | 9 | | 11 | 12 | 13 | | | | 17 | 18 | 19 | | 76.5 | 78.9 | 70.1 | 53.6 |
| 11 | 1 | 2 | | | | | | | | | 11 | 12 | 13 | 14 | | 16 | 17 | | 19 | 20 | 47.6 | 67.5 | 27.2 | 70.0 |
| 12 | 1 | 2 | 3 | | 5 | 6 | | | | | | | 13 | 14 | 15 | 16 | 17 | | | | 51.3 | 60.3 | 40.6 | 71.9 |
| 13 | | | 3 | | 5 | | 7 | 8 | 9 | | | 12 | 13 | | | 16 | | 18 | | 20 | 75.9 | 79.3 | 71.5 | 63.6 |
| 14 | 1 | 2 | 3 | 4 | | | | | | 10 | 11 | 12 | 13 | | | 16 | | | 19 | | 29.0 | 76.4 | 50.8 | 66.0 |
| 15 | | 2 | 3 | 4 | 5 | 6 | 7 | 8 | 9 | | | | | | 15 | | | | 19 | | 45.0 | 75.8 | 49.6 | 38.8 |
| 16 | 1 | 2 | | 4 | | | | | | | 11 | 12 | | 14 | | | 17 | 18 | 19 | 20 | 46.4 | 79.0 | 26.3 | 60.4 |
| 17 | 1 | 2 | | 4 | | | 7 | 8 | 9 | 10 | 11 | | | | | | | 18 | 19 | | 70.3 | 78.8 | 51.8 | 61.9 |
| 18 | | | | | 5 | 6 | 7 | 8 | 9 | 10 | | 12 | 13 | | | 16 | | 18 | | | 75.9 | 78.9 | 70.1 | 63.9 |
| 19 | | | 3 | | 5 | 6 | | | | 10 | 11 | 12 | 13 | | 15 | 16 | | 18 | | | 49.2 | 68.4 | 72.8 | 73.5 |
| 20 | 1 | 2 | 3 | | 5 | | 7 | | 9 | | 11 | | 13 | | | 16 | | | 19 | | 66.6 | 75.6 | 70.2 | 38.2 |

Table 11: **15-model combinations**. Matrix showing model presence across the 20 evaluated combinations. Values under each task report the convergence@$n$ metric computed without a credible interval; each value is the mean of $10^5$ bootstrapped samples. Model identifiers are listed in Table 7.

| Comb. | 1 | 2 | 3 | 4 | 5 | 6 | 7 | 8 | 9 | 10 | 11 | 12 | 13 | 14 | 15 | 16 | 17 | 18 | 19 | 20 | AIME'24 | AIME'25 | HMMT'25 | BrUMO'25 |
|---|---|---|---|---|---|---|---|---|---|---|---|---|---|---|---|---|---|---|---|---|---|---|---|---|
| 1 | 1 | 2 | 3 | 4 | 5 | 6 | | | | | | 12 | 13 | 14 | 15 | 16 | 17 | 18 | 19 | 20 | 59.3 | 76.1 | 72.9 | 77.1 |
| 2 | 1 | 2 | 3 | 4 | 5 | 6 | | | | 10 | 11 | | | 14 | 15 | 16 | 17 | 18 | 19 | 20 | 51.3 | 79.1 | 71.7 | 67.6 |
| 3 | | 2 | | 5 | 6 | 7 | 8 | 9 | | | | 12 | 13 | 14 | 15 | 16 | 17 | 18 | 19 | 20 | 76.6 | 79.0 | 76.4 | 77.1 |
| 4 | | 2 | 3 | | | 6 | 7 | 8 | 9 | | 11 | 12 | 13 | 14 | 15 | 16 | 17 | 18 | | 20 | 76.7 | 79.5 | 76.5 | 77.1 |
| 5 | 1 | | 3 | 4 | 5 | 6 | 7 | 8 | 9 | | | 12 | 13 | 14 | 15 | 16 | | | 19 | 20 | 68.5 | 79.6 | 71.7 | 73.9 |
| 6 | 1 | | 3 | 4 | 5 | 6 | 7 | 8 | 9 | | 11 | | | 14 | 15 | 16 | 17 | 18 | 19 | | 73.1 | 79.7 | 72.7 | 53.2 |
| 7 | 1 | 2 | | 4 | | | 7 | 8 | 9 | 10 | 11 | 12 | 13 | 14 | | | 17 | 18 | 19 | 20 | 76.5 | 79.9 | 70.3 | 69.0 |
| 8 | 1 | 2 | | 4 | 5 | | 7 | 8 | 9 | 10 | 11 | 12 | 13 | | | | 17 | 18 | 19 | 20 | 76.5 | 79.6 | 70.3 | 68.9 |
| 9 | 1 | 2 | 3 | | 5 | | 7 | 8 | 9 | 10 | 11 | 12 | 13 | | | | 17 | 18 | 19 | 20 | 76.5 | 79.3 | 71.5 | 69.9 |
| 10 | 1 | 2 | 3 | 4 | | | 7 | 8 | 9 | 10 | 11 | 12 | 13 | | | | 17 | 18 | 19 | 20 | 76.5 | 79.7 | 71.5 | 69.9 |
| 11 | | | 3 | 4 | 5 | 6 | 7 | 8 | 9 | | 11 | 12 | 13 | 14 | 15 | 16 | 17 | 18 | | | 76.7 | 79.9 | 76.5 | 75.8 |
| 12 | | | | 5 | 6 | 7 | 8 | 9 | | | 11 | 12 | 13 | 14 | 15 | 16 | 17 | 18 | 19 | 20 | 76.6 | 79.3 | 76.4 | 75.8 |
| 13 | 1 | 2 | 3 | 4 | 5 | 6 | 7 | 8 | 9 | 10 | | | | 14 | 15 | 16 | | | 19 | 20 | 54.4 | 78.7 | 55.8 | 40.3 |
| 14 | 1 | 2 | 3 | 4 | 5 | 6 | | | | 10 | 11 | 12 | 13 | 14 | 15 | 16 | | | 19 | 20 | 50.9 | 79.2 | 55.6 | 73.9 |
| 15 | 1 | 2 | 3 | 4 | 5 | 6 | | | | | 11 | 12 | 13 | 14 | 15 | 16 | 17 | 18 | 19 | | 59.4 | 79.2 | 73.0 | 77.1 |
| 16 | 1 | 2 | 3 | 4 | | 6 | | | | 10 | 11 | 12 | 13 | | 15 | 16 | 17 | 18 | 19 | 20 | 59.3 | 77.0 | 73.0 | 77.1 |
| 17 | 1 | 2 | 3 | 4 | | 6 | | | | 10 | 11 | 12 | 13 | 14 | | 16 | 17 | 18 | 19 | 20 | 52.8 | 79.2 | 57.0 | 74.9 |
| 18 | 1 | 2 | 3 | 4 | | 6 | | | | 10 | 11 | 12 | 13 | 14 | 15 | | 17 | 18 | 19 | 20 | 55.9 | 79.2 | 71.4 | 75.3 |
| 19 | 1 | 2 | 3 | 4 | 5 | 6 | 7 | 8 | 9 | 10 | | | | 14 | 15 | 16 | 17 | 18 | | | 73.1 | 78.8 | 72.7 | 66.5 |
| 20 | 1 | | 3 | 4 | 5 | 6 | 7 | 8 | 9 | | | | | 14 | 15 | 16 | 17 | 18 | 19 | 20 | 73.1 | 78.7 | 72.7 | 53.2 |

