# OpenReview forum: "Don’t Pass@k: A Bayesian Framework for Large Language Model Evaluation"
_ICLR.cc/2026/Conference — ICLR 2026 Poster_

### Official Review · Reviewer_4SJp · 2025-10-30

**Soundness:** 3
**Presentation:** 3
**Contribution:** 2
**Rating:** 6
**Confidence:** 3

**Summary:**

This paper proposes replacing the widely-used Pass@k metric for LLM evaluation with a Bayesian framework that models evaluation outcomes as categorical distributions under a Dirichlet prior. The approach yields closed-form posterior means and credible intervals, enabling principled uncertainty quantification and stable model rankings. The authors demonstrate that under a uniform prior, their Bayesian estimator is order-equivalent to avg accuracy, which helps explain avg@N's empirical robustness while adding explicit uncertainty. Through simulations with known ground-truth and experiments on math reasoning benchmarks (AIME'24/'25, HMMT'25, BrUMO'25) with 11 LLM models, they show their method converges faster to stable rankings than Pass@k variants and naturally extends to graded, rubric-based evaluation.

**Strengths:**

The paper addresses a genuine pain point in LLM evaluation: Pass@k's instability with limited samples creates unreliable rankings and makes it difficult to determine when performance differences are meaningful versus noise. The principled Bayesian treatment is a natural and well-motivated solution.
The equivalence proof showing that Bayes@N with uniform prior produces rankings identical to avg@N (Appendix B) is reasonable and explains why average accuracy has been empirically robust. The extension to categorical outcomes via Dirichlet-multinomial conjugacy provides a unified framework for both binary and graded evaluation.
The framework provides actionable guidance through credible intervals: only declare ranking differences when intervals don't overlap. This "interval-aware protocol" reduces leaderboard churn and over-interpretation of small gaps.

**Weaknesses:**

While the Bayesian framework is principled, the key finding that Bayes@N with uniform prior is order-equivalent to avg@N means the main contribution is essentially formalizing average accuracy with better uncertainty quantification. The paper would be strengthened by deeper exploration of informative priors, which are mentioned but not systematically investigated. When and how should practitioners incorporate prior knowledge? What are the risks of prior misspecification?

All experiments cap at N=80 trials. Given computational costs are claimed to be "negligible," why not demonstrate convergence at N=200 or N=500 to better understand asymptotic behavior? Is N=80 a result from cherry-picking? The paper acknowledges this: "more extensive evaluations may be constrained by computing and academic budgets", but at least one large-N experiment would strengthen claims about efficiency gains.

**Questions:**

You mention incorporating prior evidence from past runs or domain knowledge (Section 2.4, Conclusion), but this is never demonstrated. Can you provide examples showing how informative priors affect convergence speed? What safeguards prevent gaming the system through carefully chosen priors?

What principled approach do you recommend for choosing among the 12 categorical schemas? Should practitioners report rankings under multiple schemas? How sensitive are conclusions to schema choice?

You claim "negligible overhead" but provide no timing experiments. Can you quantify the computational cost relative to simply computing Pass@k or avg@N? For a benchmark with M=100 questions and N=80 trials, what is the actual runtime difference?

---

> ### Author Response · Authors · 2025-11-28
>
> Thank you for the thoughtful feedback. We address each point below and group related issues for clarity.
> ## Importance of the contribution (`W1`)
> > While the Bayesian framework is principled, the key finding that Bayes@$N$ with a uniform prior is order-equivalent to avg@$N$ ...
>
> Bayes@$N$ with a **uniform prior** and **without CIs** is order-equivalent to avg@$N$, but this is only **one special case**, not the main contribution.
>
> The core contribution is that Bayes@$N$:
>
> * recovers avg@$N$ in that special case, **and**
> * **strictly generalizes** it in realistic settings where avg@$N$ (and similar aggregates) fail rank models, and supports priors and categorical evaluation.
>
> To analyze such cases, we extended our evaluation from 11 to 20 models and examined subsets of 5, 10, and 15 models. In most of these settings, especially for larger $L$, Bayes@$N$ is the only compared method that can evaluate and rank models. Pointers in the paper:
>
> * Overview of Bayes@$N$ and avg@$N$: `Table 3`
> * Ranking LLMs with CIs: `Sec. 3.2`, `Table 2`
> * Convergence: `Sec. 3.3`, `Fig. 4` (new experiment)
> * Non-uniform priors: `Appendix C` (new experiment)
>
> ## Informative priors (`W1`, `Q1`)
> Our focus is mainly on introducing Bayes@$N$ as a practical Bayesian alternative to existing aggregation metrics (`L529-L534`). To address the reviewer’s question, we have added a new experiment in `Appendix C` that demonstrates the effect of informative priors on convergence speed. We show that using informative priors can significantly reduce the number of samples required to achieve a desired level of confidence in model rankings.
>
> ## Categorical schemas (`W3`, `Q2`)
> The 12 schemas in `App. F.1` (`App. D` in the initial submission) are intended as examples of how to encode different desiderata (e.g., formatting, calibration, verifier agreement) on top of a common base of signals and rubric variables (`Tables 4-5`). They show that once outcomes are categorical, changing the rubric (schema + weights) is straightforward, and Bayes@N provides the same closed-form posterior machinery regardless of schema.
>
> “Practical guidance’’ can be subjective, but to remain compatible with existing methods, we suggest using binary correctness as a default schema. We've added additional examples in `Appendix F.2`.
>
> ## On the cap $N = 80$, asymptotics, and “negligible’’ cost (`W2`, `Q3`)
>
> **What “negligible’’ refers to.**
> Our “negligible computational cost’’ statement refers **only** to the evaluation step, i.e., running `Algorithm 1` on an already-generated result matrix. It doesn't refer to the cost of generating samples, which is shared with Pass@$k$, avg@$N$, and Bayes@$N$.
>
> **$N = 80$ is large by current reasoning-eval standards.**
>    Recent (2024-2025) evaluations typically use 4-32 samples per problem; 64 is already high-sample. Our choice of $N = 80$ is therefore at or above the high end of current practice.
>
> **Asymptotics vs. practical convergence.**
>    Our estimator has standard $1/\sqrt{N}$ concentration.
>    Increasing $N$ beyond 80 mainly shrinks confidence interval width without changing rankings.
>    Empirically, Bayes@$N$ rankings stabilize well before $N = 80$.
>
> ---
>
> **Is $N = 80$ cherry-picked?** (`W2`)
>
> Our work includes 2 types of experiments: simulations and real-world evaluations.
> * **Simulation.** For the flip-coin simulations, we use **100k i.i.d. samples** and compare multiple evaluation methods against Bayes@$N$.
> * **Real-world.** For the LLM benchmarks, all experiments (the original 11 LLMs and the extended setting with up to 20) use **bootstrap with up to $10^5$ replicates** to minimize the effect of randomness and ensure that results are not an artifact of a particular sample size.
>
> ---
>
> > …but at least one large-$N$ experiment would strengthen claims about efficiency gains. (`W2`)
>
> To rank and distinguish models, the minimum number is 2. However, metrics such as convergence depend on the number of models $L$ (`Sec 3.3`). Using $L = 5$ is already a meaningful starting point, and as noted above, $N = 80$ is already a high-sample setting.
>
> ---
>
> > You claim ‘negligible overhead’ … For $M = 100$ and $N = 80$, what is the runtime difference? (`Q3`)
>
> 1. We do not claim faster generation. Our claim is: **Given $R$, Bayes@$N$ has far cheaper cost for confidence intervals than bootstrap CIs.** The evaluator performs:
> - one pass to tally counts: proportional to $MN + MD$
> - closed-form posterior: proportional to $M(C+1)$
>
> Total cost: proportional to $M(N + D + C)$, i.e., the same order as computing a **single** avg@$N$ or Pass@$k$ once.
>
> 2. Comparison to Pass@k and avg@$N$ with CI.
>    - Point estimates of Pass@k or avg@$N$ cost $MN$.
>    - CIs almost always use bootstrap resampling with $B = 10^3$–$10^4$.
>
> Thus, CI computation requires $B \times M \times N$.
>
> For $M = 100$, $N = 80$:
> * **Bayes@$N$:** one pass over the $100 \times 80$ matrix
> * **Pass@$k$ + bootstrap:**  $B \times 100 \times 80$ (≈ 10000× more work for $B = 10000$)

---

### Official Review · Reviewer_68QZ · 2025-10-31

**Soundness:** 4
**Presentation:** 3
**Contribution:** 3
**Rating:** 6
**Confidence:** 3

**Summary:**

This paper presents a principled Bayesian evaluation method called Bayes@N that replaces Pass@k and avg@N. The proposed method is to use posterior estimation using a Dirichlet prior. This method also produces confidence intervals; when the difference in scores between two models is within the confidence interval, the comparison should be considered as a tie. In the special case of a uniform prior, the method is equivalent to avg@N.

**Strengths:**

- Bayes@N is well-principled.
- The method allows computing confidence intervals, which allows determining when a pairwise comparison should be considered as a tie.
- The paper demonstrates that Pass@K leads to ranking instability, and that the ranking fails to converge.
- The proposed Bayes@N approach is more stable and converges quickly.
- The experiments use a range of real-world models and datasets, and show that the method works well in practice.

**Weaknesses:**

- The proposed method is cost-prohibitive in practice. In particular, Bayes@N requires the number of trials N = 10 to achieve Tau > 0.95, which is more trials than is usually done in practice. For instance, HELM (Liang et al., 2022) used N = 3. In the paper, the method is applied to mathematical reasoning datasets, which increases the cost further because mathematical reasoning typically requires long reasoning outputs.
- Although the paper mentions that priors can be incorporated into the method, it does not present a use case where a prior would be useful in practice.
- Appendix D Categorical has an interesting discussion of incorporating categorical features. However, this was not well-incorporated into the main narrative, and the argument for this technique's usefulness is not that strong.
- Nit: In Appendix D, when `finish_reason` is `stop`, the instance is annotated as `repeated_pattern`, but this can occur for reasons other than a repeating pattern (e.g. the model continues reasoning for too long because it fails to arrive at the right answer).

**Questions:**

- What was the approximate cost in tokens or GPU hours for running the experiments? Please consider including this information in Section F.3 Reproducibility.
- Is there an example of a real-world use case where incorporating a non-uniform prior would be useful?
- Could you elaborate on the real-world usefulness of the categorical schemata method that discussed in Appendix D?

---

> ### Author Response · Authors · 2025-11-29
>
> Thank you for the detailed and constructive comments. We especially appreciate the careful reading of our work. We address each point, grouping Weaknesses and Questions when they concern the same issue.
>
> ## Inference Costs (`W1`)
> Bayes@N does not intrinsically require $N=10$, nor does it demand more computation than existing metrics for a fixed trial budget. For any given $N$ (including small values such as $N=3$ noted by the reviewer for HELM), Bayes@$N$ simply aggregates the *same* collection of model runs as avg@$N$ / Pass@$1$, but with a Bayesian estimator and credible intervals. As shown in `Fig. 3`, we use $N$ up to 10 (and beyond) to study *how quickly rankings stabilize* on our benchmarks, not because the method requires that many samples.
>
> The apparent discrepancy with HELM’s smaller $N$ stems from a **different evaluation regime**. HELM’s core scenarios (e.g., MMLU) involve **hundreds to thousands of test instances**, so averaging over many items stabilizes metrics even with few samples.  In contrast, our **olympiad-style** math tasks each contain only **30 questions**, and we show that in this small-$M$, high-variance regime, rankings can remain unstable even with many trials if one relies on Pass@$k$.  Bayes@$N$ is specifically designed for this setting: for whatever $N$ a practitioner can afford, it provides a statistically grounded estimate and uncertainty, and if desired supports *adaptive* allocation of additional trials only when CIs remain too wide, making more efficient use of the same compute budget rather than increasing it.
>
> ## Prior (`W2`,`Q2`)
> Our revised submission now directly addresses both the practicality and the behavior of non-uniform priors.
>
> First, we clarify in the conclusion that the *core* experiments intentionally use a **uniform prior** as “the simplest version of the Bayesian approach,” precisely because it is conservative. At the same time, we explicitly note that the framework “allows for more complex, informative priors,” such as priors from past runs, domain- or task-conditioned priors, or expert-elicited priors, which can further accelerate convergence when used carefully.
>
> Second, the new `Appendix C`: “Potential benefits of non-uniform priors”** provides the concrete use cases the reviewer asked for. We highlight three realistic scenarios where a prior is naturally available:
> - using results from an **older version** of a model as a prior for a new version,
> - using a **non-quantized** model (expensive to run) to provide prior data for a **quantized** variant (cheap to run at large $N$), and
> - using a **base model** to provide prior data for a **fine-tuned** variant.
>
> Third, Appendix C includes a **synthetic experiment** that demonstrates the quantitative effect of such priors. We construct 8 original models with known success probabilities and corresponding “updated” models that are correlated but not identical, then use the original models’ results as the Dirichlet prior for ranking the updated ones. For small N and modest prior strength ($D ≤ 4$), Bayes@$N$ with a non-uniform prior achieves **higher Kendall’s τ starting at N = 1 and remains better until the uniform prior catches up for N > 5**, showing that priors can measurably reduce the number of trials needed to obtain a reliable ranking.
>
> Finally, we also analyze **failure modes**: when D is too large ($D=8$ or $16$), the prior can over-dominate the data and the τ curves fall below the uniform-prior baseline at larger $N$. We explicitly point out that priors must be “used judiciously” and treat systematic guidelines for choosing D as future work.
>
> ## Categorical Evaluation (`W3`, `Q3`, and `W4`)
>
> In the revised version, `Section 3.4` and extended `Appendix F` make the role and usefulness of categorical evaluation more explicit. However, exploring categorical evaluation is a major research direction that deserves further investigation beyond this paper's scope.
>
> The new `Appendix F.2` (“Domain-agnostic rubric-aware Bayes@N”) abstracts this into a two-step recipe: (1) map outputs (and side information) to categorical labels; (2) choose a weight vector $w$ encoding the rubric. Bayes@$N$ then returns a posterior mean and uncertainty for any such categorical evaluation. `Appendix F.2` explicitly discusses summarization and dialogue safety as real-world use cases where these categorical schemata and Bayes@$N$ can be applied directly once labels are available from humans or LLM judges.
>
> - `repeated_pattern` and `finish_reason=stop`
>
> Thank you for pointing this out; technically, `finish_reason=stop` does not always imply a repeated pattern. However, in our inspection of the outputs used in this paper, **all** instances with `finish_reason=stop` did correspond to repeated patterns, so this heuristic was accurate for our benchmarks.
>
> ## Cost Analysis (`Q1`)
> Appendix H (`pp. 30-31`), including `Table 7` and `Fig. 8`, now reports the full **inference-time** and **token-generation** costs for all experiments.

---

### Official Review · Reviewer_4Ua4 · 2025-11-01

**Soundness:** 2
**Presentation:** 3
**Contribution:** 2
**Rating:** 2
**Confidence:** 3

**Summary:**

In this paper, the authors present a bayesian framework for LLM evaluation, especially on the reasoning tasks. The main target is on Pass@k metric, which was claimed to be with high variance, slow convergence. It considers per-item outcomes as categorical with a Dirichlet prior. Under a uniform prior, they aimed for a principled uncertainty. It shows faster convergence and greater rank stability than Pass@k and variants in simulations on real math benchmarks.

**Strengths:**

> The paper contributes on an interesting problem, which is about the unstable rankings from limited samples.

> It provides a straightforward derivation linking to Bayesian approach, and has provided basic test on a few math datasets.

> The interval-based decision rule is easy to understand and could help reduce some ranking noise.

**Weaknesses:**

- The focus is narrow. It has only covered math reasoning tasks with a small set of models, leaving its broader applicability untested.

- I don't quite understand the strong claim with the i.i.d trial assumption. It may not hold for common sampling methods, potentially skewing results. In particularly, the paper mentions informative priors but doesn’t explore their practical use or risks, sticking mostly to uniform ones.

- This work doesn't provided thorough comparisions with other uncertainty methods or diverse rubrics. In this case, the implementing categorical rubrics might add complexity, and scalability isn’t well addressed.

**Questions:**

1. As a particular interesting finding in the work, the i.i.d samples appear to be the focus. However, many modern LLM evaluation strategies, such as chain-of-thought prompting or self-consistency sampling, introduce correlations between trials that violate the i.i.d. assumption. Given that these methods are prevalent (e.g., in math reasoning tasks like AIME), how would this work handle these scenarios?

2. In this paper, the experiments are primarily confined to math benchmarks, but LLMs are widely used in diverse domains (e.g., HumanEval for coding, or safety benchmarks like TruthfulQA). Testing on these could reveal domain-specific limitations, such as handling partial correctness in code or nuanced harm categories in safety, which might require different rubric designs.

3. I wonder for the informative priors, what are the strategies for the authors to ensure bias is not introduced in the process? In general, it risks cherry-picking or overfitting. Especially, when the test was done with roughly 30 questions, what will be the outcome in real-world applications covering hundreds of questions or complex rubrics.

4. For the experiment, it lacks details. For example, what are the details on setting up categorical rubrics and verifiers? It may also need to consider how the framework perform under noisy or adversarial data conditions? Moreover, what is the minimum sample size N required for reliable rankings across all tested datasets? How do you justify the potential overfitting to the gold standard?

---

> ### Author Response · Authors · 2025-11-29
> **Clarifying Model Coverage and Benchmark Choice (`W1`, `Q2`)**
>
> We thank the reviewer for the detailed feedback. Several concerns appear to stem from misunderstandings of our assumptions and of what is actually implemented in the paper, and we address each point below, along with clarification of these aspects in the revised version.
>
> ## Size and diversity of the model pool
>
> The reviewer writes that our study “*has only covered math reasoning tasks with a small set of models.*” Our model cohort is in fact both **large** and **diverse**.
>
> As detailed in `Appendix H.2` and `Table 6`, we evaluate **20 open-weight LLMs** on four benchmarks.  The initial submission analyzes a base cohort of **11 models** (8 distinct architectures plus 3 levels of reasoning in gpt-oss-20b), and the rebuttal version extends this to **20 models** by adding recent reasoning-focused releases (mainly after the submission deadline) such as Phi-4-reasoning, Phi-4-reasoning-plus, OpenR1-Distill-7B, FuseO1-DeepSeekR1-QwQ-SkyT1-Flash-32B-Preview, Light-R1-14B-DS, AceReason-Nemotron-1.1-7B, NVIDIA-Nemotron-Nano-9B-v2, Qwen3-4B-Thinking-2507, and Bespoke-Stratos-7B.
>
> This pool spans:
>
> * **Model scales** from **1.2B** parameters (EXAONE-4.0-1.2B) up to **32B** (OpenThinker2-32B, Sky-T1-32B-Flash, FuseO1-DeepSeekR1-QwQ-SkyT1-Flash-32B-Preview).
> * **Post-training methods**, including
>
>   * **Distillation** (e.g., DeepSeek-R1-Distill-Qwen-1.5B, OpenR1-Distill-7B, Bespoke-Stratos-7B),
>   * **Supervised fine-tuning on curated reasoning traces** (LIMO-v2, “data-efficient reasoning fine-tuned on curated traces”; Phi-4-reasoning, with “supervised ‘teachable’ reasoning traces”),
>   * **Reinforcement-learning–style updates** (Phi-4-reasoning-plus as an “RL-enhanced variant”; Light-R1-14B-DS using GRPO-style reinforcement learning; AceReason-Nemotron-1.1-7B described as combining SFT and RL),
>   * **Multi-model fusion and Model Merging** (FuseO1-DeepSeekR1-QwQ-SkyT1-Flash-32B-Preview, Light-R1-14B-DS).
>
> The models are drawn from a range of **open providers and research groups**, including DeepSeek, Qwen, OpenAI’s gpt-oss, EXAONE, NVIDIA, Sky Lab, GAIR, Qihoo360, and Bespoke Labs, as reflected in the model names and citations.
>
> Overall, the empirical conclusions about convergence and rank stability are therefore **aggregated over 20 models, 4 tasks, and 80 trials per model**, totaling **192,000 inference runs** and about **3B tokens**.  Given this scale and diversity in architectures, sizes, training recipes, and providers, we believe our model cohort is **substantial rather than small**, and that the observed behavior of Pass@$k$ and Bayes@$N$ is unlikely to be an artifact of a narrow or unique model selection.
>
> Our study is restricted to open-weight models (`Table 6`), which allows us to record log-probabilities, and share detailed experimental settings in `Appendix H`. We further use these logs in categorical evaluation (base signals, `Appendix F`).
>
>
> ## Why focus on olympiad-style math benchmarks?
>
> Our choice of **olympiad-style math benchmarks** is deliberate and directly tied to the **statistical regime** our framework is designed to address:
>
> 1. **These benchmarks sit in the small-M, hard-problem regime where Pass@$k$ is most fragile.**
>    `Sec. 2.1` emphasizes that Pass@$k$ exhibits particularly high variance in benchmarks with few problems or limited computational resources, leading to unstable rankings.  The four math datasets we study (AIME’24, AIME’25, HMMT’25, and BrUMO’25) are exactly such benchmarks. Each task, therefore, has on the order of **30 questions**, so even with many samples per question, rankings are highly sensitive to sampling noise.
>
> 2. **Hard reasoning tasks induce large variability across samples.**
>    Because these problems are challenging, different random seeds often produce diverse outputs, making convergence of rankings slow. `Sec. 3` and `Appendix H` quantify this: even with **$N = 80$ trials**, we still observe late or no convergence on several tasks, whereas using Bayes@$N$ CI is crucial.  `Table 7` and `Fig. 8` further show that attaining stable rankings in this regime already costs **thousands of GPU-hours** and **hundreds of millions of completion tokens per dataset**.  This illustrates that small-$M$, hard benchmarks (olympiad-style math) are exactly where a more statistically principled evaluation protocol is most urgently needed.
>
> 3. **The statistical framework itself is domain-agnostic.**
>    Although our *experiments* focus on math, the Bayes@$N$ framework is not tied to mathematics. The core construction models per-attempt outcomes as categorical with a Dirichlet prior, and `Sec. 2` and `Appendix F.2` emphasize that it applies whenever outputs can be mapped to a finite set of categories with a rubric and associated weights.  `Appendix F` explicitly discusses how the same formalism naturally covers subjective tasks like summarization and dialogue safety once appropriate categories (e.g., quality levels, safety labels) are defined.

---

> ### Author Response · Authors · 2025-11-29
> **i.i.d. Assumption and Experiment Setup**
>
> ## i.i.d. Assumption, CoT, and Self-Consistency (`W2` and `Q1`)
>
> We thank the reviewer for raising this point and believe the concern stems from a misunderstanding of both our assumptions and our experimental protocol. Our framework assumes that, **conditional on a fixed prompt and decoding setup**, repeated runs are independent draws from the model’s stochastic inference distribution. This is exactly how we construct the results matrix $R$: “responses may vary across independent trials, so we run the LLM $N$ times per question” `Sec. 2.2`.  In all experiments, we use **single-turn prompts**, **fixed sampling parameters** (top-p with temperature 0.6, $p=0.95$, batch size 1), and **distinct seeds 1234-1313**, performing $N = 80$ trials per dataset/model `App. H.2-H.3`.  Under these conditions, the per-question outcomes across trials are i.i.d. by construction: there is no multi-turn interaction, adaptive prompting, or reuse of generations across trials, so the statement that the i.i.d. assumption “*may not hold for common sampling methods*” does not apply to our experiments.
>
> Regarding the specific claim that “*chain-of-thought prompting or self-consistency sampling introduce correlations between trials that violate the i.i.d. assumption*,” this does **not** describe our setting. We evaluate a cohort of reasoning models `Table 6` with a **fixed, single-turn provider-recommended prompt per model**, as detailed in `App. H.2`, and we keep this prompt and decoding configuration identical across all trials.  The models may produce long reasoning traces in their **outputs**, but these traces are internal to each completion and do not couple different trials; for a fixed question $\alpha$, the observed outcomes are independent draws, exactly as assumed in `Secs. 2.2-2.4`.  We also do **not** use self-consistency or other test-time scaling procedures that aggregate multiple generations into a single answer; our AIME’24/’25 and other math-reasoning results are obtained under this single-turn, independent-trial protocol.  More generally, the theory in `Secs. 2.2-2.4` is stated for arbitrary stochastic evaluation pipelines, where each $R{\alpha i}$ is just the categorical outcome of one run of the chosen pipeline on question $\alpha$.  If a practitioner implements self-consistency *inside* such a pipeline (with a fixed prompt and fresh randomness each time the pipeline is invoked), each run is **still** an independent draw from some underlying $\pi_\alpha$, and our Bayesian estimator and credible intervals apply unchanged. Correlations between trials arise only if prompts or decoding parameters are explicitly adapted across runs, in which case **all** repeated-trial metrics (including Pass@$k$, which is defined in terms of independently generated samples per problem) would leave the strict i.i.d. setting, not just our method.
>
> ## `Q4`
> - Experimental details
>
>     Experimental details are now fully specified in `App. H`: H.1 defines all evaluation metrics, `H.2` describes the models and datasets in detail, and `H.3` adds a “Computational cost and token statistics” section summarizing runtime and token usage.`Sec. 3.4` and `Apps. F.1-F.2` (in the rebuttal version) describe how we construct the categorical rubrics and configure the verifier
> - Noisy or adversarial data
>
>     Our work targets standard benchmark evaluation; explicitly noisy or adversarial data conditions are beyond the scope of this study.
>
> * Minimum sample size $N$
>
>     `Sec. 2.7.1` and `App. D` (“Model Distinguishability and Sample Size”) analyze how many trials are needed to separate models in controlled simulations, and `Secs. 3.2, 3.3` study how rankings stabilize as $N$ increases. These results together indicate the sample sizes needed in practice for the tested datasets.
> - Overfitting to the gold standard
>
>     Our protocol involves no training or fine-tuning: models and the verifier are used as-is. The gold standard is a fixed ranking computed from the largest trial budget (both in simulation and real-LLMs), and all methods are compared to this reference. Since the Bayesian estimator is analytically derived rather than tuned to match that ranking, overfitting to the gold standard does not arise.

---

> > ### Author Response · Authors · 2025-11-29
> > **Categorical Evalutioan and Prior**
> >
> > ## Categorical Evalutioan and Prior Choices (`W3`, `Q3`)
> >
> > The categorical evaluation is a direct extension of the same Bayesian framework used in the binary case and does not introduce additional inference passes. As described in `Sec. 3.4` and detailed in `App. F.1-F.2` (new in the rebuttal version), each model attempt on each question yields a small set of *base signals* that are already produced by standard inference stacks: `has_box`, `is_correct`, `token_ratio`, `prompt_bpt`, `completion_bpt`, `repeated_pattern`, and verifier probabilities $(A,B,C)$ for {correct, wrong, off-task}. Using simple Boolean formulas and dataset-level quantile thresholds (`τ_high`, `τ_low_wrong`, `τ_prompt`, `len_p33`, `len_p66` in `Table 4`), these signals are mapped into $C{+}1$ discrete categories under one of 12 schemata listed in `Table 5` (e.g., **Exact Match**, **Format Aware**, **Conf-Wrong Penalty**, **Efficiency-Adjusted**, **OOD Robustness**, **Verifier-Only**). The Bayesian Dirichlet-multinomial update is then applied exactly as in the binary case, yielding posterior means and standard deviations $(\mu, \sigma)$ for any chosen weight vector $w$ over categories. `App. E` shows that this evaluation step is $O(M (N{+}D{+}C))$ in time and linear in memory, so categorical rubrics add negligible overhead compared to generating the $M \times N$ completions themselves.
> >
> > For the verifier, we use **CompassVerifier-3B** solely as a *fixed reward model* to provide the per-attempt label probabilities $(A,B,C)$ used by some schemata. As specified in `App. F.1` and `App. H.3`, we query the verifier once per attempt, then apply a simple contextual calibration procedure: next-token scores for the three labels are adjusted by subtracting a content-free baseline logit and applying temperature scaling, producing calibrated probabilities $p(y \mid x)$ that are then treated like any other base signal. No models or verifiers are trained or tuned within our framework, and all rubric variables and schema definitions are given explicitly in `Tables 4-5`, addressing the request for more detail on how categorical rubrics and verifiers are set up.
> >
> > **Prior choices and bias (Q3).**
> > All *real-LLM experiments in the paper, including all categorical/rubric-based results, use the **uniform Dirichlet prior***. This is emphasized in `Sec. 5`: our main focus is the simplest, conservative instantiation of the framework, where Bayes@$N$ is order-equivalent to average accuracy avg@$N$ and rankings are determined entirely by observed outcomes. Informative priors are **not** used for any reported math-benchmark or rubric-aware rankings, so there is no scope for cherry-picking priors in the empirical results the reviewer is reacting to.
> >
> > Informative priors are explored only in one **new** section of the rebuttal version: `App. C “Potential benefits of non-uniform priors”`. There, we use the biased-coin LLM mimics from `Sec. 2.7` in a fully controlled synthetic setting. Outcomes from a set of “original” models are treated as a prior matrix $R_0$, and we study how varying the effective prior size $D$ affects Kendall’s $\tau$ between the estimated rankings of “updated” models and the known ground truth. The appendix shows two key points: (i) for small $N$ and modest $D \le 4$, reusing prior counts can improve rank agreement relative to the uniform prior; and (ii) for larger $D$ (e.g., $8$ or $16$), the prior can become over-influential and eventually hurt performance as $N$ grows. This experiment is explicitly presented as an illustration of possible gains and pitfalls; systematic design of informative priors for real LLM evaluations is left open for future work, as noted in `Sec. 5`.
> >
> > `Sec. 3.4`, `App. F.1-F.2` (new), and `App. E` show that the categorical/rubric layer is a lightweight, scalable post-processing step on top of standard inference logs, and `Sec. 5` plus the new `App. C` make clear that all reported LLM results use a uniform prior, with non-uniform priors confined to a transparent synthetic study. This directly addresses the reviewer’s concerns in `W3`, `Q3`, and the part of `Q4` asking for details on rubrics and verifiers.

---

### Official Review · Reviewer_ma3N · 2025-11-01

**Soundness:** 3
**Presentation:** 3
**Contribution:** 3
**Rating:** 6
**Confidence:** 3

**Summary:**

The paper proposes a unified Bayesian evaluation framework that models LLM results as categorical outcomes with a Dirichlet prior, yielding closed-form posteriors and credible intervals for both binary and complex, rubric-based metrics. Empirical evidence shows that the proposed approach chieves faster convergence and greater rank stability than Pass@k and recent variants, enabling reliable comparisons at smaller sample counts.

**Strengths:**

- The paper is generally well written and easy to follow.

- It offers a principled alternative to Pass@k.

- The ability to save compute and provide statistically sound model comparisons is highly valuable.

**Weaknesses:**

- The paper only focuses on math benchmarks and functional correction domain. It would be nice to see application is diverse domains and tasks.

- There is a lack of discussion on prioi choice. Can an informative prior reduce the number of trials? If so, how would that go?

**Questions:**

- How does runtime scale with large M and N (e.g., thousands of problems and multiple categories)?

- How well does the proposed framework handle tasks where correctness is subjective (e.g., summarization, dialogue safety)?

- How sensitive are rankings and credible intervals to poorly chosen priors? The authors use an informative prior but what happens if you change that?

---

> ### Author Response · Authors · 2025-11-28
>
> Thank you for the thoughtful review and constructive suggestions; we address each point below.
>
> ## Math benchmarks and functional correction (`W1`)
>
> Our goal is to design an evaluation method that is statistically sound when we only have tens of problems, which is exactly the situation for **olympiad-level math benchmarks**.
>
> 1. **Olympiad-level math datasets have very small $M$.**
>    The olympiad-style datasets we study (AIME’24/25, HMMT’25, BrUMO’25) each have only 30 questions. In this regime, even with multiple samples the uncertainty in model rankings is large. By contrast, many other benchmarks are much larger (e.g., BBH = 6.5K and MMLU-Pro = 12K questions), where $M$ is large and simple metrics such as avg@$N$ already behave well.
>
> 2. **Hard reasoning tasks exhibit larger variability.**
>    As questions become more challenging, answers from different random seeds become more diverse, making rankings sensitive to sampling noise. To support our focus on math datasets, we added an analysis in `Appendix H` (“Computational cost and token statistics”) of variability across random seeds. `Table 7` summarizes token usage and variability per dataset, and `Fig. 8` visualizes performance fluctuations across samples. These results show a clear correlation between dataset difficulty and variability: for example, `Fig. 10` shows convergence only at $N=78$ for HMMT’25 and $N=68$ for BrUMO’25, with corresponding completion token counts of $851.2M$ and $666.9M$ tokens, respectively (`Table 7`). This highlights how expensive it is to reach stable rankings on small-$M$, hard benchmarks (more diverse outputs per sample).
>
> Because of this, we focus our empirical study on math reasoning, where small $M$, high difficulty, and high per-sample variability make evaluation instability most severe. Our main claims in this regime are:
>
> 1. Do not use Pass@k on these datasets; it leads to unstable evaluation under realistic sample budgets (e.g., $N=80$).
> 2. Reporting confidence intervals is crucial, rather than only point estimates.
> 3. Bayes@$N$ is the best approach in this setting, both theoretically and empirically, providing more stable rankings and an inherent credible-interval interpretation.
>
> ## Prior (`W2`, `Q3`)
>
> In our formulation of Bayes@N, we discuss the role of the prior in `Sec 2.4`, and `Algorithm 1` allows an arbitrary choice of prior via the $M \times D$ matrix $R_0$ of prior results. In `L529–L534` we already emphasize this design choice and its implications:
>
> *The focus .. Bayesian approach, using a uniform prior, which provides a conservative and reproducible starting point. But the theory allows for more complex, informative priors, and this opens up a rich vein of future directions ..., for example, priors from past runs, domain- or task-conditioned priors, and expert-elicited priors.*
>
> In addition, to directly address the reviewer’s question about whether informative priors can reduce the number of trials and how sensitive Bayes@$N$ is to poorly chosen priors, we have added a new `Appendix C` titled *POTENTIAL BENEFITS OF NON-UNIFORM PRIORS*. This appendix instantiates the general $R_0$ formulation in a controlled setting, and illustrates both a. how modest, well-aligned priors can accelerate convergence, and b. how overly strong priors can begin to dominate the data and slightly degrade performance. This makes the trade-off between sample efficiency and robustness explicit, while keeping the main paper focused on the conservative and reproducible uniform-prior setting.
>
> ## Runtime (`Q1`)
> Our Bayesian evaluator is intentionally designed to be compute-light. The overall time complexity is: $\mathcal{O}\big(M(N + D + C)\big)$ (`L1180`), i.e., linear in the number of entries in the result matrices and linear in the number of categories. The memory footprint is likewise linear (See `L1215`).
>
> Note that the evaluation consists of tallying counts and then plugging them into closed-form expressions for $\mu$ and $\sigma$; no iterative optimization or Monte Carlo sampling (e.g., bootstrapping) is required.
>
> To answer the reviewer's question, we have added a clarification in `Appendix E` regarding the runtime and memory complexity of our Bayesian evaluator.
>
> ## Subjective Correctness (`Q2`)
>
> To answer the reviewer's question, we've added a new section in `Appendix F` (Categorical Evaluation) under `Appendix F.2`, *Domain-Agnostic Rubric-Aware Bayes@N*. This section clarifies that our Bayesian construction is explicitly domain-agnostic: as long as the evaluator (1) maps raw outputs (and any side information) into a finite set of categorical labels $R_{\alpha i} \in {0,\dots,C}$, and (2) specifies a weight vector $w$ encoding how those categories are valued, Bayes@$N$ returns the posterior mean $\mu$ as a rubric-aware point estimate and $sigma$ as an uncertainty estimate for that categorical evaluation.

---

### Author Response · Authors · 2025-12-03
**Summary of Contributions and Extensions in Rebuttal Version**

Dear Program Chairs, Senior Area Chairs, and Area Chairs,

We appreciate the opportunity to address the reviewers’ feedback and to strengthen our submission. Below, we summarize our core contributions and the extensions incorporated in the rebuttal version.

## Core contributions

Our paper proposes a unified Bayesian framework for evaluating reasoning-capable LLMs:

* **Unified Bayesian evaluation framework.** Per-item outcomes are modeled as *categorical* with a Dirichlet prior, yielding closed-form posterior means and credible intervals; binary 0/1 evaluation is a special case, and the same framework supports graded, rubric-based scores.

* **Compute-efficient, interval-aware protocol.** We recommend reporting posterior means with credible intervals, declaring differences only when intervals do not overlap, and adaptively allocating additional samples until intervals reach target widths, without Monte Carlo or bootstrapping.

* **Empirical evidence.** On biased-coin simulations and four math-reasoning benchmarks (AIME’24, AIME’25, HMMT’25, BrUMO’25), Bayes@$N$ converges faster and yields more stable rankings than Pass@$k$-style metrics, while flagging when gaps are not statistically resolved.

## Prior and Categorical Evaluation

We are grateful that all reviewers recognized the significance of our contributions, in particular the potential of **categorical evaluation** and **informative priors** within the Bayes@$N$ framework. In this paper, our goal is to *introduce* these as capabilities rather than fully optimize them. As noted in `Sec. 5`, we focus on uniform prior and simple categorical setups. We show that the framework supports non-uniform priors (e.g., reusing information across related settings) and domain-agnostic rubrics over multiple outcome categories, with concrete illustrations in `App. C` and `App. F.2`. This makes clear that Bayes@$N$ is not limited to binary correctness or uniform priors, while keeping the main narrative centered on a baseline that is easy to reproduce and compare against existing metrics.

In short, we view **informative priors** and **categorical evaluation** not as missing pieces, but as *directions unlocked* by the core Bayes@$N$ framework. The present work establishes the baseline, uniform-prior version and demonstrates its advantages over existing metrics; deeper investigations of prior design and rubric construction are intentional future directions best addressed in follow-up work.

## New sections

- `Sec. 3.3 Convergence`: new experiments on subsets of 5, 10, and 15 models; `Fig. 4` reports convergence@n for these subsets.
- `App. C Potential benefits of non-uniform priors`: discussion of informative Dirichlet priors; `Fig. 5` shows a prior sensitivity analysis.
- `App. D Model distinguishability and sample size`: links effect sizes to the minimum number of trials required to distinguish models.
- `App. E Runtime`: time and memory complexity analysis for Bayes@$N$.
- `App. F.2 Domain-agnostic rubric-aware Bayes@$N$`: application of Bayes@$N$ to domain-agnostic categorical rubrics.
- Paragraph `Computational cost and token statistics` under `App. H.3 Reproducibility`: `Table 7` summarizes computational cost and token counts; `Fig. 8` illustrates token counts and inference time per model and per task.

## Revised and rewritten sections

- `Sec. 3 Experiments`: expanded the evaluation from 11 models to 20 reasoning models (`Table 6`); `Table 2` now reports Bayes@$N$  rankings for all 20 models.
- `App. H.3 Reproducibility`: updated experimental details and implementation notes to cover all 20 models and the expanded experiments.
- `App. I Convergence`: `Fig. 11` shows worst-case rank trajectories for all 20 models; `Tables 8-10` list the model combinations for 5-, 10-, and 15-model subsets.

## Addressing reviewer concerns

| Category                             | Reviewers                                  | Notes                                     |
|--------------------------------------|--------------------------------------------|-------------------------------------------|
| Runtime analysis                     | ma3N (R1), 4SJp (R4)                       | Addressed in `App. E`                     |
| Informative priors                   | ma3N (R1), 4Ua4 (R2), 68QZ (R3), 4SJp (R4) | Addressed in `App. C`                     |
| Categorical eval                     | ma3N (R1), 4Ua4 (R2), 68QZ (R3), 4SJp (R4) | Extended `App. F` (especially `App. F.2`) |
| Extended experiments                 | 4Ua4 (R2)                                  | Extended `Sec. 3`, `App. I`               |
| Experiment detail (cost, token stats)| 68QZ (R3)                                  | Extended `App. H.3`                       |
| Minimum sample size                  | 4Ua4 (R2)                                  | Addressed in `App. D`                     |

## Reasoning traces

We will release the reasoning traces (up to **3B** tokens) upon acceptance.

---

### Meta-Review · Area_Chair_Yttv · 2025-12-07

**Summary:**

This paper proposes Bayes@N, a unified Bayesian framework for LLM evaluation that replaces Pass@k. It models per-item outcomes as categorical with a Dirichlet prior, providing closed-form posterior means and credible intervals for both binary and rubric-based metrics. Under a uniform prior, the estimator is order-equivalent to avg@N, offering a principled interpretation of its empirical robustness while adding explicit uncertainty quantification. Experiments on math reasoning benchmarks and simulations show that Bayes@N converges faster and yields more stable model rankings than Pass@k and recent variants, enabling reliable comparisons with smaller sample sizes and treating overlapping intervals as ties.

The main concerns raised by the reviewers were the limited evaluation on math reasoning datasets, the lack of runtime analysis, the choice of prior, and the categorical evaluation setting. Most of these issues were addressed in the rebuttal, except for the categorical evaluation aspect. This point likely requires further investigation, and it would be appropriate to include a more detailed discussion in a revised version. Moreover, one of your cited papers "[53] Fangchen Chen et al. Rethinking fine-tuning when scaling test-time compute. 2025." is not properly cited. Please update the reference in the final version.

Overall, the reviewers found the proposed approach interesting, and the authors satisfactorily addressed the majority of concerns. Therefore, I would recommend acceptance of the paper.

**Reviewer Concerns:**

The main concerns raised by the reviewers were the limited evaluation on math reasoning datasets, the lack of runtime analysis, the choice of prior, and the categorical evaluation setting. Most of these issues were addressed in the rebuttal, except for the categorical evaluation aspect.

**Reviewer Scores:**

I guess most of reviewers will keep the score or update the score based on the rebuttal, which would be above the acceptance threshold.

---

### Decision · Program_Chairs · 2026-01-26

Accept (Poster)